# BIOSCAN-5M: A Multimodal Dataset for Insect Biodiversity

**Zahra Gharaee**[3*], **Scott C. Lowe**[5*], **ZeMing Gong**[4*], **Pablo Millan Arias**[3*],
**Nicholas Pellegrino**[3], **Austin T. Wang**[4], **Joakim Bruslund Haurum**[7],
**Iuliia Zarubiieva**[2,5], **Lila Kari**[3],
**Dirk Steinke**[1,2†], **Graham W. Taylor**[2,5†], **Paul Fieguth**[3†], **Angel X. Chang**[4,6†]

[1]Centre for Biodiversity Genomics, [2]University of Guelph, [3]University of Waterloo,
[4]Simon Fraser University, [5]Vector Institute, [6]Alberta Machine Intelligence Institute (Amii),
[7]Aalborg University and Pioneer Centre for AI
https://biodiversitygenomics.net/5M-insects/

## Abstract

As part of an ongoing worldwide effort to comprehend and monitor insect biodiversity, this paper presents the BIOSCAN-5M Insect dataset to the machine learning community and establish several benchmark tasks. BIOSCAN-5M is a comprehensive dataset containing multi-modal information for over 5 million insect specimens, and it significantly expands existing image-based biological datasets by including taxonomic labels, raw nucleotide barcode sequences, assigned barcode index numbers, geographical, and size information. We propose three benchmark experiments to demonstrate the impact of the multi-modal data types on the classification and clustering accuracy. First, we pretrain a masked language model on the DNA barcode sequences of the BIOSCAN-5M dataset, and demonstrate the impact of using this large reference library on species- and genus-level classification performance. Second, we propose a zero-shot transfer learning task applied to images and DNA barcodes to cluster feature embeddings obtained from self-supervised learning, to investigate whether meaningful clusters can be derived from these representation embeddings. Third, we benchmark multi-modality by performing contrastive learning on DNA barcodes, image data, and taxonomic information. This yields a general shared embedding space enabling taxonomic classification using multiple types of information and modalities. The code repository of the BIOSCAN-5M Insect dataset is available at https://github.com/bioscan-ml/BIOSCAN-5M.

## 1 Introduction

Biodiversity plays a multifaceted role in sustaining ecosystems and supporting human well-being. Primarily, it serves as a cornerstone for ecosystem stability and resilience, providing a natural defence against disturbances such as climate change and invasive species (Cardinale et al., 2012). Additionally, biodiversity serves as a vital resource for the economy, supplying essentials like food, medicine, and genetic material (Sala et al., 2000). Understanding biodiversity is paramount for sustainable resource management, ensuring the availability of these resources for future generations (Duraiappah et al., 2005). To understand and monitor biodiversity, Gharaee et al. (2023) introduced the BIOSCAN-1M Insect dataset, which pairs DNA with images, as a stepping stone to developing AI tools for automatic classification of organisms.

---

*Joint first author. †Joint senior/last author.

38th Conference on Neural Information Processing Systems (NeurIPS 2024) Track on Datasets and Benchmarks.

Figure 1: The BIOSCAN-5M Dataset provides taxonomic labels, a DNA barcode sequence, barcode index number, a high-resolution image along with its cropped and resized versions, as well as size and geographic information for each sample.

However, that work only investigated image classification down to the family level, focusing on the Diptera order, and did not fully utilize the multimodal nature of the dataset. In addition, BIOSCAN-1M was limited to specimen collected from just 3 countries and the *Insecta* `class`. Expanding upon BIOSCAN-1M, we introduce the BIOSCAN-5M dataset—a comprehensive repository of multi-modal information (see Figure 1) on over 5 million arthropod specimens (98% insects), with 1.2 million labelled to `genus` or `species` taxonomic ranks. Compared to its predecessor, the BIOSCAN-5M dataset offers a significantly larger volume of high-resolution microscope images and DNA barcodes along with critical annotations, including taxonomic ranks, size, and geographical information. Additionally, we performed data cleaning to resolve inconsistencies and provide more reliable labels.

The multimodal characteristics of BIOSCAN-5M are not only essential for biodiversity studies, but also facilitate further innovation in machine learning and AI. In this paper, we conduct experiments that leverage the multimodal aspects of BIOSCAN-5M, extending its application beyond the image-only modality used in Gharaee et al. (2023). Here, we train the masked language model (MLM) proposed in BarcodeBERT (Millan Arias et al., 2023) on the DNA barcodes of the BIOSCAN-5M dataset and demonstrate the impact of using this large reference library on species- and genus-level classification. We achieve an accuracy higher than that of state-of-the-art models pretrained on more general genomic datasets, especially in the 1NN-probing task of assigning samples from unseen species to seen genera. Next, we perform a zero-shot transfer learning task (Lowe et al., 2024a) through zero-shot clustering representation embeddings obtained from encoders trained with self-supervised paradigms. This approach demonstrates the effectiveness of pretrained embeddings in clustering data, even in the absence of ground-truth. Finally, as in CLIBD (Gong et al., 2024), we learn a shared embedding space across three modalities in the dataset—high-quality RGB images, textual taxonomic labels, and DNA barcodes—for fine-grained taxonomic classification.

## 2    Related work

### 2.1    Datasets for taxonomic classification

Biological datasets are essential for advancing our understanding of the natural world, with uses in genomics (Network et al., 2013), proteomics (Kim et al., 2014), ecology (Kattge et al., 2011), evolutionary biology (Flicek et al., 2014), medicine (Jensen et al., 2012), and agriculture (Lu & Young, 2020; Xu et al., 2023; Galloway et al., 2017; He et al., 2024). Table 1 compares biological datasets used for taxonomic classification. Many of these datasets feature fine-grained classes and exhibit a long-tailed class distribution, making the recognition task challenging for machine learning (ML) methods that do not account for these properties. While many datasets provide images, they do not include other attributes such as DNA barcode, or geographical locations. Most relevant to our work is BIOSCAN-1M Insect (Gharaee et al., 2023), which introduced a dataset of 1.1 M insect images paired with DNA barcodes and taxonomic labels.

DNA barcodes are short, highly descriptive DNA fragments that encode sufficient information for species-level identification. For example, a DNA barcode of an organism from Kingdom Animalia (Hebert et al., 2003; Braukmann et al., 2019) is a specific 648 bp sequence of the cytochrome c oxidase I (COI) gene from the mitochondrial genome, used to classify unknown individuals and discover new species (Moritz & Cicero, 2004). DNA barcodes have been successfully applied to taxonomic identification and classification, ecology, conservation, diet analysis, and food safety (Ruppert et al., 2019; Stoeck et al., 2018), offering faster and more accurate results than traditional meth-

Table 1: **Summary of fine-grained and long-tailed biological datasets.** The "Taxa" column indicates the taxonomic scope of each dataset. The "IR" column is the class imbalance ratio, computed as the ratio of the number of samples in the largest category to the smallest category.

| Dataset | Reference | Year | Images | IR | Taxa | Rank | Categories | Taxon | BIN | DNA | Geography | Size |
|---|---|---|---|---|---|---|---|---|---|---|---|---|
| LeafSnap | Kumar et al. (2012) | 2012 | 31 k | 8 | Plants | Species | 184 | ✗ | ✗ | ✗ | ✗ | ✗ |
| NA Birds | Van Horn et al. (2015) | 2015 | 48 k | 15 | Birds | Species | 400 | ✗ | ✗ | ✗ | ✗ | ✗ |
| Urban Trees | Wegner et al. (2016) | 2016 | 80 k | 7 | Trees | Species | 18 | ✗ | ✗ | ✗ | ✗ | ✗ |
| DeepWeeds | Olsen et al. (2019) | 2019 | 17 k | 9 | Plants | Species | 9 | ✗ | ✗ | ✗ | ✓ | ✗ |
| IP102 | Wu et al. (2019) | 2019 | 75 k | 14 | Insects | Species | 102 | ✓ | ✗ | ✗ | ✗ | ✗ |
| Pest24 | Wang et al. (2020) | 2020 | 25 k | 494 | Insects | Species | 24 | ✗ | ✗ | ✗ | ✗ | ✗ |
| Pl@ntNet-300K | Garcin et al. (2021) | 2021 | 306 k | 3,604 | Plants | Species | 1,000 | ✗ | ✗ | ✗ | ✗ | ✗ |
| iNaturalist (2021) | Van Horn et al. (2021) | 2021 | 2,686 k | 2 | All | Species | 10,000 | ✓ | ✗ | ✗ | ✗ | ✗ |
| iNaturalist-Insect | Van Horn et al. (2021) | 2021 | 663 k | 2 | Insects | Species | 2,526 | ✓ | ✗ | ✗ | ✗ | ✗ |
| Species196-L | He et al. (2024) | 2023 | 19 k | 351 | Various | Mixed | 196 | ✓ | ✗ | ✗ | ✗ | ✗ |
| CWD30 | Ilyas et al. (2023) | 2023 | 219 k | 61 | Plants | Species | 30 | ✓ | ✗ | ✗ | ✗ | ✗ |
| BenthicNet | Lowe et al. (2024b) | 2024 | 1,429 k | 22,394 | Aquatic | Mixed | 791 | ✓ | ✗ | ✗ | ✓ | ✗ |
| Insect-1M | Nguyen et al. (2024b) | 2024 | 1,017 k | N/A | Arthropods | Species | 34,212 | ✓ | ✗ | ✗ | ✗ | ✗ |
| BIOSCAN-1M | Gharaee et al. (2023) | 2023 | 1,128 k | 12,491 | Insects | BIN* | 90,918 | ✓ | ✓ | ✓ | ✗ | ✗ |
| **BIOSCAN-5M** | Ours | 2024 | 5,150 k | 35,458 | Arthropods | BIN* | 324,411 | ✓ | ✓ | ✓ | ✓ | ✓ |

* For datasets that include Barcode Index Numbers (BINs) annotations, we present BINs, which serve as a (sub)species proxy for organisms and offer a viable alternative to Linnean taxonomy.

ods (Pawlowski et al., 2018). Barcodes can also be grouped together based on sequence similarity into clusters called Operational Taxonomic Units (OTUs) (Sokal & Sneath, 1963; Blaxter et al., 2005), each assigned a Barcode Index Number (BIN) (Ratnasingham & Hebert, 2013). In general, biological datasets may also incorporate other data such as labels for multi-level taxonomic ranks, which can offer valuable insights into the evolutionary relationships between organisms. However, datasets with hierarchical taxonomic annotations (He et al., 2024; Ilyas et al., 2023; Liu et al., 2021; Wu et al., 2019; Gharaee et al., 2023) are relatively scarce.

## 2.2 Self-supervised learning

Self-supervised learning (SSL) has recently gained significant attention for its ability to leverage vast amounts of unlabelled data, producing versatile feature embeddings for various tasks (Balestriero et al., 2023). This has driven the development of large-scale language models (Brown et al., 2020) and computer vision systems trained on billions of images (Goyal et al., 2021). Advances in transformers pretrained with SSL at scale, known as foundation models (Ji et al., 2021; Zhou et al., 2023; Dalla-Torre et al., 2023; Zhou et al., 2024; Chia et al., 2022; Gu et al., 2021), have shown robust performance across diverse tasks.

Recent work has leveraged these advances for taxonomic classification using DNA. Since the introduction of the first DNA language model, DNABERT (Ji et al., 2021), which mainly focused on human data, multiple models with different architectures and tokenization strategies have emerged (Mock et al., 2022; Zhou et al., 2023, 2024; Millan Arias et al., 2023; Nguyen et al., 2024a) with some incorporating data from multiple species during pretraining and allowing for species classification (Zhou et al., 2023, 2024; Millan Arias et al., 2023). These models are pretrained to be task-agnostic, and are expected to perform well after fine-tuning in downstream tasks. Yet, their potential application for taxonomic identification of arbitrary DNA sequences or DNA barcodes has not been extensively explored. One relevant approach, BERTax (Mock et al., 2022), pretrained a BERT (Dosovitskiy et al., 2021b) model for hierarchical taxonomic classification on broader ranks such as kingdom, phylum, and genus. For DNA barcodes specifically, BarcodeBERT (Millan Arias et al., 2023) was developed for species-level classification of insects, with assignment to genus for unknown species.

Although embeddings from SSL-trained feature extractors exhibit strong performance on downstream tasks post fine-tuning, their utility without fine-tuning remains underexplored. Previous studies (Vaze et al., 2022; Zhou & Zhang, 2022) suggest that SSL feature encoders produce embeddings conducive to clustering, albeit typically after fine-tuning. A recent study (Lowe et al., 2024a) has delved into whether SSL-trained feature encoders *without* fine-tuning can serve as the foundation for clustering, yielding informative clusters of embeddings on real-world datasets unseen during encoder training.

## 2.3 Multimodal Learning

There has been a growing interest in exploring multiple data modalities for biological tasks (Ikezogwo et al., 2024; Lu et al., 2023; Zhang et al., 2023). Badirli et al. (2021) introduced a Bayesian zero-shot

learning approach, leveraging DNA data to model priors for species classification based on images. Those authors also employed Bayesian techniques (Badirli et al., 2023), combining image and DNA embeddings in a unified space to predict the genus of unseen species.

Recent advances in machine learning allowed scalable integration of information across modalities. For example, CLIP (Radford et al., 2021) used contrastive learning to encode text captions and images into a unified space for zero-shot classification. BioCLIP (Stevens et al., 2024) used a similar idea to align images of organisms with their common names and taxonomic descriptions across a dataset of 10 M specimens encompassing plants, animals, and fungi. CLIBD (Gong et al., 2024) used a contrastive loss to align the three modalities of RGB images, textual taxonomic labels, and DNA barcodes. By aligning these modalities, CLIBD can use either images or DNA barcodes for taxonomic classification and learn from incomplete taxonomic labels, making it more flexible than BioCLIP (Stevens et al., 2024), which requires full taxonomic annotations for each specimen.

## 3   Dataset

The BIOSCAN-5M dataset is derived from Steinke et al. (2024) and comprises 5,150,850 `arthropod` specimens, with insects accounting for about 98% of the total. The diverse features of this dataset are described in this section. BIOSCAN-5M is a superset of the BIOSCAN-1M Insect dataset (Gharaee et al., 2023), providing more samples and additional metadata such as geographical location.

**Images.** The BIOSCAN-5M dataset provides specimen images at 1024×768 pixels, captured using a Keyence VHX-7000 microscope. Figure 2 showcases the diversity in organism morphology across the dataset. The images are accessed via the `processid` field of the metadata as {`processid`}.jpg. Following BIOSCAN-1M Insect (Gharaee et al., 2023), the images are cropped and resized to 341×256 pixels to facilitate model training. We fine-tuned DETR (End-to-End Object Detection with Transformers) for image cropping. For BIOSCAN-1M Insect, the cropping model was trained using 2 k insect images. Building on the BIOSCAN-1M Insect cropping tool checkpoint, we fine-tuned the model for BIOSCAN-5M using the same 2 k images and an additional 837 images that were not well-cropped previously. This fine-tuning process followed the same training setup, including batch size, learning rate, and other hyper parameter settings (see supplement for details). The bounding box of the cropped region is provided as part of the dataset release.

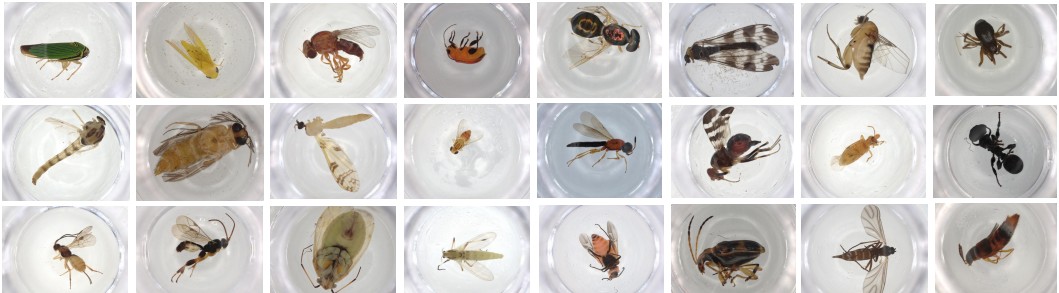

Figure 2: Samples of original full-size images of distinct organisms in the BIOSCAN-5M dataset.

**Genetic-based indexing.** The genetic information of the BIOSCAN-5M dataset described in §2 is represented as the raw nucleotide barcode sequence, under the `dna_barcode` field, and the Barcode Index Number under `dna_bin` field. Independently, the field `processid` is a unique number assigned by BOLD (International Barcode of Life Consortium, 2024) to each record, and `sampleid` is an identifier given by the collector.

**Biological taxonomic classification.** Linnaean taxonomy is a hierarchical classification system instigated by Linnaeus (1758) for organizing living organisms which has been developed over several hundred years. It categorizes `species` based on shared characteristics and establishes a standardized naming convention. The hierarchy includes several taxonomic ranks, such as `domain`, `kingdom`, `phylum`, `class`, `order`, `family`, `genus`, and `species`, allowing for a structured approach to studying biodiversity and understanding the relationships between different organisms.

The dataset samples undergo taxonomic classification using a hybrid approach involving an AI-assisted tool proposed by Gharaee et al. (2023) and human taxonomic experts. After DNA barcoding and sequence alignment, the taxonomic levels derived from both the AI tool and DNA sequencing are compared. Any discrepancies are then reviewed by human experts. Importantly, assignments to deeper taxonomic levels, such as `family` or lower, rely entirely on human expertise. Labels at seven taxonomic ranks are used to represent individual specimens, denoted by fields `phylum`, `class`, `order`, `family`, `subfamily`, `genus`, and `species`.

Table 2: Summary statistics of dataset records by taxonomic rank.

| Attributes | BIOSCAN-5M (Ours) | | | | BIOSCAN-1M (Gharaee et al., 2023) | | |
| | IR | Categories | Labelled | Labelled (%) | Categories | Labelled | Labelled (%) |
|---|---|---|---|---|---|---|---|
| phylum | 1 | 1 | 5,150,850 | 100.0 | 1 | 1,128,313 | 100.0 |
| class | 719,831 | 10 | 5,146,837 | 99.9 | 1 | 1,128,313 | 100.0 |
| order | 3,675,317 | 55 | 5,134,987 | 99.7 | 16 | 1,128,313 | 100.0 |
| family | 938,928 | 934 | 4,932,774 | 95.8 | 491 | 1,112,968 | 98.6 |
| subfamily | 323,146 | 1,542 | 1,472,548 | 28.6 | 760 | 265,492 | 23.5 |
| genus | 200,268 | 7,605 | 1,226,765 | 23.8 | 3,441 | 254,096 | 22.5 |
| species | 7,694 | 22,622 | 473,094 | 9.2 | 8,355 | 84,397 | 7.5 |
| dna_bin | 35,458 | 324,411 | 5,137,441 | 99.7 | 91,918 | 1,128,313 | 100.0 |
| dna_barcode | 3,743 | 2,486,492 | 5,150,850 | 100.0 | 552,629 | 1,128,313 | 100.0 |

In the source data, we found identical DNA nucleotide sequences labelled differently at some taxonomic levels, which was likely due to human error (e.g. typos) or disagreements in the taxonomic labelling. To address this, we checked and cleaned the taxonomic labels to address typos and ensure consistency across DNA barcodes (see supplement for details). We note that some of the noisy species labels are placeholder labels that do not correspond to well-established scientific taxonomic species names. In our data, the placeholder `species` labels are identified by `species` labels that begin with a lowercase letter, contain a period, contain numerals, or contain "malaise".

Statistics for BIOSCAN-5M are given in Table 2 for the seven taxonomic ranks along with the BIN and DNA nucleotide barcode sequences. For each group, we report the number of categories, and the count and fraction labelled. We compute the class imbalance ratio (IR) as the ratio of the number of samples in the largest category to the smallest category, reflecting the class distribution within each group. For more detailed statistical analysis, see the supplementary materials.

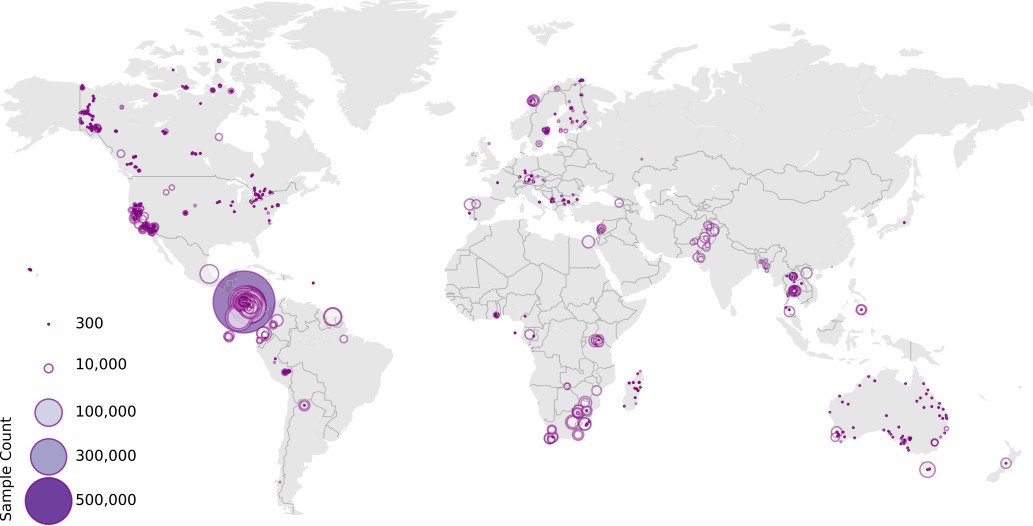

Figure 3: Geographical locations obtained from latitude and longitude coordinates of the regions where the samples of the BIOSCAN-5M dataset were collected.

**Geographic location.** The BIOSCAN-5M dataset includes geographic location information, detailing the country and province or state where each specimen is collected, along with the latitude and longitude coordinates of each collection site. This information is detailed in the fields `country`, `province_state`, `coord-lat` and `coord-lon`. The distribution of specimen collection sites are shown on a world map in Figure 3.

**Challenges.** The BIOSCAN-5M dataset faces two key challenges: First, there exists a sampling bias as a result of the locations where and the methods through which specimens are collected. Second, the number of labelled records sharply declines at deeper taxonomic levels, especially beyond the family rank, which makes fine-grained classification tasks more challenging.

# 4 Benchmark experiments and results

In real-world insect biodiversity monitoring, it is common to encounter both species which are already known to science, and samples whose species is novel. Thus, to excel in biodiversity monitoring, a model must correctly categorize instances of known species, and identify novel species outside the existing taxonomy, grouping together samples of the same new species. In our experiments, we explore three methods which offer utility in these regards, evaluated in two settings: closed-world and open-world. In the closed-world setting, the task is to accurately identify species from a predefined set of existing labels. In the open-world setting, the task is to group together samples of novel species.

## 4.1 Data partitioning

**Species sets.** We first partition records based on their species label into one of four categories, with all samples bearing the same species label being placed in the same species set. *Seen*: all samples whose species label is an established scientific name of a species. *Unseen*: labelled with an established scientific name for the genus, and a uniquely identifying placeholder name for the species. *Heldout*: labelled with a placeholder genus and species name. *Unknown*: samples without a species label (note: these may truly belong in any of the other three categories).

Table 3: Statistics and purpose of our data partitions.

| Species set | Split | Purpose | # Samples | # Barcodes | # Species |
|---|---|---|---|---|---|
| unknown | `pretrain` | self- and semi-sup. training | 4,677,756 | 2,284,232 | — |
| seen | `train` | supervision; retrieval keys | 289,203 | 118,051 | 11,846 |
| | `val` | model dev; retrieval queries | 14,757 | 6,588 | 3,378 |
| | `test` | final eval; retrieval queries | 39,373 | 18,362 | 3,483 |
| unseen | `key_unseen` | retrieval keys | 36,465 | 12,166 | 914 |
| | `val_unseen` | model dev; retrieval queries | 8,819 | 2,442 | 903 |
| | `test_unseen` | final eval; retrieval queries | 7,887 | 3,401 | 880 |
| heldout | `other_heldout` | novelty detector training | 76,590 | 41,250 | 9,862 |

**Splits.** Using the above species sets, we establish partitions for our experiments (Table 3). The *unknown* samples are all placed into a `pretrain` split for use in self-supervised pretraining and/or semi-supervised learning. As some DNA barcodes are common to multiple samples, for *seen* and *unseen* records we split the records by placing all samples with the same barcode in the same partition, to ensure there is no repetition of barcodes across splits. For the closed-world setting, we use the *seen* records to establish `train`, `val`, `test` splits. To ensure that the `test` set is not too imbalanced in species distribution, we place samples in the `test` set with a flattened distribution. We sample records from species with at least two unique barcodes and eight samples, and the number of samples placed in the `test` set scales linearly with the total number of samples for the species, until reaching a cap of 25 samples. We sample 5% of the remaining *seen* data to form the `val` partition, but in this case match the imbalance of the overall dataset. The remaining samples then form the `train` split, with a final split distribution of $84.2 : 4.3 : 11.5$. Following standard practice, the `val` set is for model evaluation during development and hyperparameter tuning, and the `test` set is for final evaluation. In the retrieval setting, the `train` split should additionally be used as a database of *keys* to retrieve over, and the `val` and `test` split as queries. For additional details on the partitioning method and statistical comparisons between the partitions, please see the supplementary materials.

For the open-world scenario, we use a similar procedure to establish `val_unseen` and `test_unseen` over the *unseen* records. After creating `test_unseen` with the same methodology as `test`, we sample 20% of remaining *unseen* species records to create `val_unseen`. The remaining *unseen* species samples form the `keys_unseen` set. In the retrieval setting, `keys_unseen` is used to form the database of *keys* to retrieve, and the `val_unseen` and `test_unseen` splits act as queries. The *heldout* species samples form a final `other_heldout` partition. As these species are in neither *seen* nor *unseen*, this split can be used to train a novelty detector without using any *unseen* species.

## 4.2 DNA-based taxonomic classification

In this section, we demonstrate the utility of the BIOSCAN-5M dataset for DNA-based taxonomic classification. Due to their standardized length, DNA barcodes are ideal candidates as input to CNN- and transformer-based architectures for supervised taxonomic classification. However, as noted by Millan Arias et al. (2023), a limitation of this approach is the uncertainty in species-level labels for a substantial portion of the data. This uncertainty, partly due to the lack of consensus among researchers and the continuous discovery of new species, may render supervised learning suboptimal for this task. We address this issue by adopting a semi-supervised learning approach. Specifically, we train a model using self-supervision on unlabelled sequences from the `pretrain` split and the `other_heldout` split, followed by fine-tuning on sequences from the `train` split, which includes high-quality labels. The same pretrained model can be used to produce embeddings for sequences from unseen taxa to address tasks in the open-world setting. Consequently, we use these embeddings to perform non-parametric taxonomic classification at a higher (less specific) level in the taxonomic hierarchy for evaluation.

**Experimental setup.** Although there has been a growing number of SSL DNA language models proposed in the recent literature, the results obtained by the recently proposed BarcodeBERT (Millan Arias et al., 2023) model empirically demonstrate that training on a dataset of DNA barcodes can outperform more sophisticated training schemes that use a diverse set of non-barcode DNA sequences, such as DNABERT (Ji et al., 2021) and DNABERT-2 (Zhou et al., 2023). In this study, we selected BarcodeBERT as our reference model upon which to investigate the impact of pretraining on the larger and more diverse DNA barcode dataset BIOSCAN-5M. See Appendix A for pretraining details.

We compare our pretrained model against four pretrained transformer models: BarcodeBERT (Millan Arias et al., 2023), DNABERT-2 (Zhou et al., 2023), DNABERT-S (Zhou et al., 2024), and the nucleotide transformer (NT) (Dalla-Torre et al., 2023); one state space model, HyenaDNA (Nguyen et al., 2024a); and a CNN baseline following the architecture introduced by Badirli et al. (2021).

As an additional assessment of the impact of BIOSCAN-5M DNA data during pretraining, we use the different pretrained models as feature extractors and evaluate the quality of the embeddings produced by the models on two different SSL evaluation strategies (Balestriero et al., 2023). We first implement genus-level 1-NN probing on sequences from unseen species, providing insights into the models' abilities to generalize to new taxonomic groups. Finally, we perform species-level classification using a linear classifier trained on embeddings from the pretrained models. Note that for both probing tasks, all the embeddings produced by a single sequence are averaged across the token dimension to generate a token embedding for the barcode.

**Results.** We leverage the different partitions of the data and make a distinction between the experiments in the closed-world and open-world settings. In the closed-world setting, the task is species-level identification of samples from species that have been seen during training (Fine-tuned accuracy, Linear probing accuracy). For reference, BLAST (Altschul et al., 1990), an algorithmic sequence alignment tool, achieves an accuracy of 99.78% in the task (not included in Table 4 as it is not a machine learning model). In fine-tuning, our pretrained model with a 8-4-4 architecture achieves the highest accuracy with 99.28%, while DNABERT-2 achieves 99.23%, showing competitive performance. Overall, all models demonstrate strong performance in this task, showcasing the effectiveness of DNA barcodes in species-level identification. For linear probing accuracy, DNABERT-S outperforms others with 95.50%, followed by our model (8-4-4) with 94.47%. BarcodeBERT ($k$=4) and DNABERT-S also show strong performance with 91.93% and 91.59% respectively (see Table 4).

In the open-world setting, the task is to assign samples from unseen species to seen categories of a coarser taxonomic ranking (1NN-genus probing). In this task, BLAST achieves an accuracy of 58.74% (not in the table), and our model (8-4-4) performs notably well with an 47.03% accuracy, which is

Table 4: **Performance of DNA-based sequence models** in closed- and open-world settings. For the closed-world, we show the species-level accuracy (%) for predicting seen species (`test`), for open-world the genus-level accuracy (%) for `test_unseen` species while using seen species to fit the model. Bold indicates highest accuracy, underlined denotes second highest.

| Model | Architecture | SSL-Pretraining | Tokens seen | Seen: Species | | Unseen: Genus |
| | | | | Fine-tuned | Linear probe | 1NN-Probe |
|---|---|---|---|---|---|---|
| CNN baseline | CNN | – | – | 97.70 | – | 29.88 |
| NT | Transformer | Multi-Species | 300 B | 98.99 | 52.41 | 21.67 |
| DNABERT-2 | Transformer | Multi-Species | 512 B | **99.23** | 67.81 | 17.99 |
| DNABERT-S | Transformer | Multi-Species | ~1,000 B | 98.99 | **95.50** | 17.70 |
| HyenaDNA | SSM | Human DNA | 5 B | 98.71 | 54.82 | 19.26 |
| BarcodeBERT | Transformer | DNA barcodes | 5 B | 98.52 | 91.93 | 23.15 |
| Ours (8-4-4) | Transformer | DNA barcodes | 7 B | **99.28** | 94.47 | **47.03** |

significantly higher than the other transformer models. The CNN baseline and HyenaDNA show lower accuracies of 29.88% and 19.26%, respectively. The use of DNA barcodes for pretraining in our models and BarcodeBERT demonstrates effectiveness in both seen and unseen species classification tasks. One limitation of the comparison is the difference in the dimension of the output space of the different models (128 for HyenaDNA, *vs.* 512 for NT and 768 for the BERT-based models). The selection of our model (8-4-4) as the best-performing configuration was done after performing a hyperparameter search to determine the optimal value of $k$ for tokenization, as well as the optimal number of heads and layers in the transformer model. To do that, after pretraining, we fine-tuned the model for species-level identification and performed linear- and 1NN- probing on the `validation` split (see Table 6). We finally note that our pretrained model outperforms BarcodeBERT, the other model trained exclusively trained on DNA barcodes, across all tasks.

### 4.3 Zero-shot transfer-learning

Recently, Lowe et al. (2024a) proposed the task of *zero-shot clustering*, investigating how well unseen datasets can be clustered using embeddings from pretrained feature extractors. Lowe et al. (2024a) found that BIOSCAN-1M images were best clustered taxonomically at the family rank while retaining high clustering performance at species and BIN labels. We replicate this analysis using BIOSCAN-5M and extend the modality space to include both image and DNA barcodes.

**Experimental setup.** We follow the experimental setup of Lowe et al. (2024a). (1) Take pretrained encoders; (2) Extract feature vectors from the stimuli by passing them through an encoder; (3) Reduce dimensions to 50 using UMAP (McInnes et al., 2018); (4) Cluster the reduced embeddings with Agglomerative Clustering (L2, Ward's method) (Everitt et al., 2011); (5) Evaluate against the ground-truth annotations with Adjusted Mutual Information (AMI) score (Vinh et al., 2010), measuring the percentage information explained relative to the entropy of the true labels.

For the image encoders, we consider ResNet-50 (He et al., 2016) and ViT-B (Dosovitskiy et al., 2021a) models, each pretrained on ImageNet-1K (Russakovsky et al., 2015) using either cross-entropy supervision (X-ent.), or SSL methods (MAE: He et al., 2022; VICReg: Bardes et al., 2022; DINO-v1: Caron et al., 2021; MoCo-v3: Chen et al., 2021). We also considered the CLIP (Radford et al., 2021) encoder, which was pretrained on an unspecified, large dataset of captioned images. To cluster the DNA barcodes, we used recent pretrained models (see §4.2 and Appendix A.2), which feature a variety of model architectures, pretraining datasets, and training methodologies: BarcodeBERT (Millan Arias et al., 2023), DNABERT-2 (Zhou et al., 2023), DNABERT-S (Zhou et al., 2024), the nucleotide transformer (NT) (Dalla-Torre et al., 2023), and HyenaDNA (Nguyen et al., 2024a).

We only cluster samples from the `test` and `test_unseen` splits. None of the image or DNA pretraining datasets overlap with BIOSCAN-5M, so all samples are "unseen". However, we note that there is a greater domain shift from the image pretraining datasets than the DNA pretraining datasets.

**Results.** Similar to Lowe et al. (2024a), we find (Figure 4) image clusterings agree with the taxonomic labels at coarse ranks (order: 88%), but agreement decreases progressively at finer-grained ranks (species: 21%); the best-performing image encoder was DINO, followed by other SSL methods VICReg and MoCo-v3, with (larger) ViT-B models outperforming ResNet-50 models. We found the

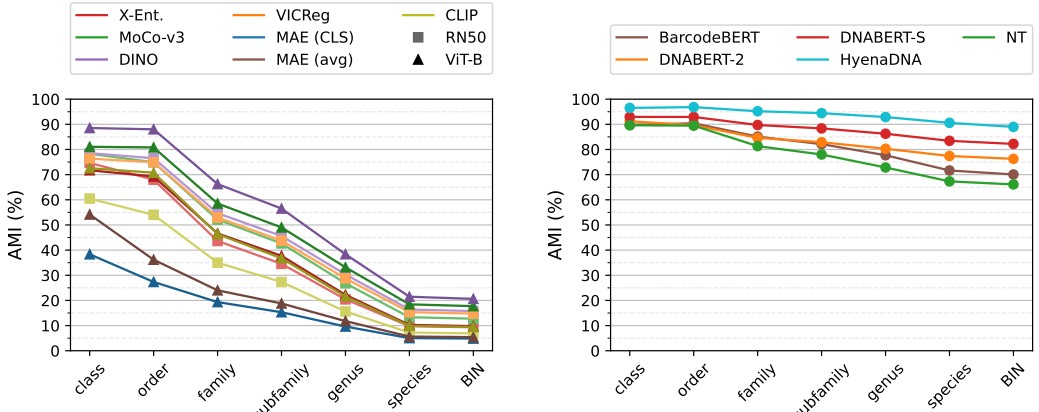

Figure 4: **Zero-shot clustering AMI (%) performance** across taxonomic ranks. Left: Image encoders. Right: DNA encoders.

performance of the DNA encoders exceeded that of the image encoders across all taxonomic levels, with higher performance at coarse ranks (order: 97%) and much shallower decline as granularity becomes finer (species: 91%). HyenaDNA provided the best performance, with 90% agreement between its clusterings and both the GT species and DNA BIN annotations. These results suggest that DNA barcodes are highly informative about species identity (which is unsurprising as it is the reason this barcode is used), and unseen samples can be readily grouped together using off-the-shelf DNA models.

We also considered the zero-shot clustering of the concatenated image and DNA representations, detailed in Appendix B.3. Due to the high performance of the DNA features, adding image features to the embeddings decreased the performance compared to using DNA embeddings alone. For additional details and analysis, see Appendix B.

### 4.4 Multimodal retrieval learning

Lastly, we demonstrate the importance of a multimodal dataset through alignment of image, DNA, and taxonomic label embeddings using CLIBD (Gong et al., 2024) to improve taxonomic classification. By learning a shared embedding space across modalities, we can query between modalities and leverage the information across them to achieve better performance in downstream tasks. We are able to incorporate a diversity of samples into training toward taxonomic classification, even with incomplete taxonomic labels.

**Experimental setup.** We follow the model architecture and experimental setup of CLIBD (Gong et al., 2024). We start with pretrained encoders for each modality and perform full-tuning with NT-Xent loss (Sohn, 2016). Our image encoder is a ViT-B (Dosovitskiy et al., 2021a) pretrained on ImageNet-21k and fine-tuned on ImageNet-1k (Deng et al., 2009). For DNA barcodes, we use BarcodeBERT (Millan Arias et al., 2023) with 5-mer tokenization, pretrained on 893 k DNA barcodes from the Barcode of Life Data system (BOLD) (International Barcode of Life Consortium, 2024), and for text, we use BERT-small (Turc et al., 2019). We train on our `pretrain` and `train` splits using the Adam (Kingma & Ba, 2014) optimizer for 20 epochs until convergence with a learning rate of 1e-6, batch size 2000. Training took 29 hours on four 80GB A100 GPUs. To evaluate the performance of our models, we report micro (see Appendix C) and macro top-1 accuracy for taxonomic classification at different levels. To determine the taxonomic labels for a new query, we encode the sample image or DNA and find the closest matching embedding in a set of labelled samples (keys). For efficient lookup, we use FAISS (Johnson et al., 2019) with exact search (`IndexFlatIP`).

We compared the performance for the initial pretrained (unimodal) encoders to our models fine-tuned on either the full `pretrain` and `train` partitions from BIOSCAN-5M, or on a random 1 million sample subset of these partitions. The 1M image subset contained 20% of the images, 27% of the barcodes, and 47% of the BINs of the 5 M image training dataset. We evaluated these using image-to-image, DNA-to-DNA, and image-to-DNA embeddings as queries and keys.

Table 5: Top-1 macro accuracy (%) on the test set for using different amount of pre-training data (1 million vs 5 million records from BIOSCAN-5M) and different combinations of aligned embeddings (image, DNA, text) during contrastive training. We show results for using image-to-image, DNA-to-DNA, and image-to-DNA query and key combinations. As a baseline, we show the results prior to contrastive learning (no alignment). We report the accuracy for seen and unseen species, and the harmonic mean (H.M.) between these (bold: highest acc.).

| Taxon | # Records | Aligned embeddings | | | DNA-to-DNA | | | Image-to-Image | | | Image-to-DNA | | |
|---|---|---|---|---|---|---|---|---|---|---|---|---|---|
| | | Img | DNA | Txt | Seen | Unseen | H.M. | Seen | Unseen | H.M. | Seen | Unseen | H.M. |
| Order | — | ✗ | ✗ | ✗ | 95.8 | 97.8 | 96.8 | 78.1 | 82.4 | 80.2 | 3.6 | 6.3 | 4.6 |
| | 1M | ✓ | ✓ | ✓ | **100.0** | **100.0** | **100.0** | 93.5 | 95.6 | 94.5 | 86.5 | 95.4 | 90.7 |
| | 5M | ✓ | ✓ | ✗ | **100.0** | **100.0** | **100.0** | **95.3** | 96.2 | 95.7 | **92.2** | 97.2 | **94.7** |
| | 5M | ✓ | ✓ | ✓ | **100.0** | **100.0** | **100.0** | 95.1 | **98.5** | 96.8 | 91.2 | **98.0** | 94.4 |
| Family | — | ✗ | ✗ | ✗ | 90.2 | 92.1 | 91.2 | 52.3 | 55.5 | 53.8 | 0.3 | 1.0 | 0.4 |
| | 1M | ✓ | ✓ | ✓ | 98.3 | 99.3 | 98.8 | 86.8 | 89.9 | 88.3 | 65.8 | 73.7 | 69.5 |
| | 5M | ✓ | ✓ | ✗ | 99.4 | **100.0** | **99.7** | 91.0 | 92.7 | 91.8 | 80.5 | 83.6 | 82.0 |
| | 5M | ✓ | ✓ | ✓ | **99.5** | **100.0** | **99.7** | **91.7** | **94.2** | **93.0** | **80.9** | **84.6** | **82.7** |
| Genus | — | ✗ | ✗ | ✗ | 86.8 | 85.7 | 86.2 | 34.0 | 31.9 | 32.9 | 0.0 | 0.0 | 0.0 |
| | 1M | ✓ | ✓ | ✓ | 98.0 | 97.2 | 97.6 | 76.5 | 75.6 | 76.1 | 46.2 | 36.2 | 40.6 |
| | 5M | ✓ | ✓ | ✗ | **99.0** | 99.3 | **99.2** | 83.3 | 85.5 | 84.4 | **64.4** | 50.4 | **56.6** |
| | 5M | ✓ | ✓ | ✓ | 98.8 | **99.5** | **99.2** | **84.0** | **86.0** | **85.0** | 63.0 | **50.6** | 56.1 |
| Species | — | ✗ | ✗ | ✗ | 84.6 | 75.6 | 79.8 | 24.2 | 12.6 | 16.6 | 0.0 | 0.0 | 0.0 |
| | 1M | ✓ | ✓ | ✓ | 96.7 | 91.7 | 94.1 | 66.6 | 49.6 | 56.8 | 34.9 | 6.8 | 11.3 |
| | 5M | ✓ | ✓ | ✗ | **98.1** | 95.8 | **97.0** | 75.9 | **60.8** | **67.5** | **54.4** | **13.8** | **22.0** |
| | 5M | ✓ | ✓ | ✓ | 98.0 | **95.9** | **97.0** | **76.0** | 60.1 | 67.1 | 51.1 | 12.7 | 20.3 |

**Results.** We compare CLIBD trained on the full BIOSCAN-5M training set against models trained on a randomly selected subset of 1 million records and the initial pretrained encoders before multimodal contrastive learning. Our results, shown in Table 5, demonstrate that our full model improves classification accuracy for same-modality queries and enables cross-modality queries. By aligning to DNA, our image embeddings are able to capture finer details. We likewise see improvements in alignment among DNA embeddings. Additionally, we observe that increasing the training dataset size from 1 million to 5 million records leads to better models with more accurate results across all studied taxa for both image and DNA modalities, indicating there are still benefits from dataset scale at this size. By including the text modality, we further improve accuracy at the higher taxa levels. Interestingly, including the text modality results in slightly lower performance at the species level. This is likely due to the sparse availability of species labels in the training data, as only 9% of records having species labels. For additional details and analysis, see Appendix C.

## 5  Conclusion

We present the BIOSCAN-5M dataset, a valuable resource for the machine learning community containing over 5 million arthropod specimens. To highlight the dataset's multimodal capabilities, we conducted three benchmark experiments that leverage images, DNA barcodes, and textual taxonomic annotations for fine-grained taxonomic classification and zero-shot clustering.

An open problem for biodiversity monitoring systems is handling novel species. To facilitate research in this space, our dataset includes partitions for both closed-world and open-world settings. Furthermore, we provide three distinct benchmark tasks, each evaluated down to species-level, demonstrating the real-world applicability of BIOSCAN-5M's multimodal features. These tasks include fine-grained taxonomic classification using DNA sequences, multimodal classification combining DNA, images, and taxonomic labels, and clustering of learned DNA and image embeddings.

We believe that the BIOSCAN-5M dataset will serve as a catalyst for further machine learning research in biodiversity, fostering innovations that can enhance our understanding and preservation of the natural world. By providing a curated multi-modal resource, we aim to support further initiatives in the spirit of TreeOfLife-10M (Stevens et al., 2024) and contribute to the broader goal of mapping and preserving global biodiversity. This dataset not only facilitates advanced computational approaches but also underscores the crucial intersection between technology and conservation science, driving forward efforts to protect our planet's diverse ecosystems for future generations.

## Acknowledgments and Disclosure of Funding

We acknowledge the support of the Government of Canada's New Frontiers in Research Fund (NFRF), [NFRFT-2020-00073]. This research is also supported by an NVIDIA Academic Grant. This research was enabled in part by support provided by Calcul Québec[1] and the Digital Research Alliance of Canada[2]. Resources used in preparing this research were provided, in part, by the Province of Ontario, the Government of Canada through CIFAR, and companies sponsoring[3] the Vector Institute. Data collection was enabled by funds from the Walder Foundation, a New Frontiers in Research Fund (NFRF) Transformation grant, a Canada Foundation for Innovation's (CFI) Major Science Initiatives (MSI) Fund and CFREF funds to the Food from Thought program at the University of Guelph. The authors also wish to acknowledge the team at the Centre for Biodiversity Genomics responsible for preparing, imaging, and sequencing specimens used for this study. We also thank Mrinal Goshalia for assistance with the cropping tool and annotation of images.

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

# Appendices

## A  DNA-based Taxonomic Classification — Additional Experiments

As described in the main text (§4.2), we leverage all data splits in the BIOSCAN-5M dataset by adopting a semi-supervised learning approach. Specifically, we train a model using self-supervision on the unlabelled partition of the data, followed by fine-tuning on the train split. Our experimental setup is illustrated in Figure 5.

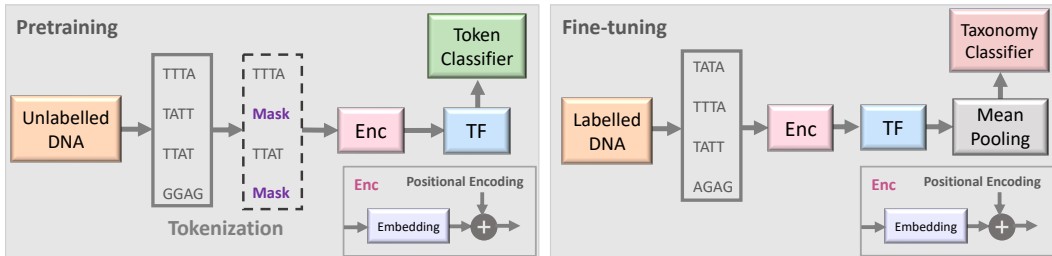

Figure 5: **DNA-based taxonomic classification methodology.** Two stages of the proposed semi-supervised learning set-up based on BarcodeBERT (Millan Arias et al., 2023). (1) Pretraining: DNA sequences are tokenized using non-overlapping $k$-mers and 50% of the tokens are masked for the MLM task. Tokens are encoded and fed into a transformer model. The output embeddings are used for token-level classification. (2) Fine-tuning: All DNA sequences in a dataset are tokenized using non-overlapping $k$-mer tokenization and all tokenized sequences, without masking, are passed through the pretrained transformer model. Global mean-pooling is applied over the token-level embeddings and the output is used for taxonomic classification.

### A.1  Pretraining details

We pretrain the model on the 2,283,900 unique DNA sequences from the `pretrained` partition and the 41,232 unique sequences from the `other_heldout` partition, totalling 2,325,132 pretraining DNA samples. For all samples, trailing `N` characters are removed and all sequences are truncated at 660 nucleotides. Note that leading `N` characters are retained since they are likely to correspond to true unknown nucleotides in the barcode. The model was pretrained using the same MLM loss function and training configurations as in BarcodeBERT (Millan Arias et al., 2023). Specifically, we use a non-overlapping $k$-mer-based tokenizer and a transformer model with 12 transformer layers, each having 12 attention heads. However, we included a random offset of at most $k$ nucleotides to each sequence as a data augmentation technique to enhance the sample efficiency. We use a learning rate of $2 \times 10^{-4}$, a batch size of 128, a OneCycle scheduler (Smith & Topin, 2017), and the AdamW optimizer (Loshchilov & Hutter, 2019), training the model for 35 epochs. In addition to using the architecture reported in BarcodeBERT, we performed a parameter search to determine the optimal $k$-mer tokenization length and model size, parameterized by the number of layers and heads in the transformer model, in order to identify an optimal architecture configuration. After pretraining, we fine-tuned the model with cross-entropy supervision for species-level classification. The pre-training stage takes approximately 50 hours using four Nvidia A40 GPUs and the fine-tuning stage of the 4-12-12 models takes 2.5 hours in four Nvidia A40 GPUs.

### A.2  Baseline Models

There has been a growing number of SSL DNA language models proposed in recent literature, most of which are based on the transformer architecture and trained using the MLM objective (Ji et al., 2021; Zhou et al., 2023, 2024). These models differ in the details of their model architecture, tokenization strategies, and training data but the underlying principles remain somewhat constant. An exception to this trend is the HyenaDNA (Nguyen et al., 2024a) model, which stands out by its use of a state space model (SSM) based on the Hyena architecture (Poli et al., 2023) and trained for next token prediction. For evaluation, we utilized the respective pre-trained models from Huggingface ModelHub, specifically:

- DNABERT-2: zhihan1996/DNABERT-2-117M
- DNABERT-S: zhihan1996/DNABERT-S
- NT: InstaDeepAI/nucleotide-transformer-v2-50m-multi-species
- HyenaDNA: LongSafari/hyenadna-tiny-1k-seqlen

The BarcodeBERT implementation was taken from `https://github.com/Kari-Genomics-Lab/BarcodeBERT`. All the models, including our pretrained models, were fine-tuned for 35 epochs with a batch size of 32 or 128 and a learning rate of $1 \times 10^{-4}$ per 64 samples in the batch with the OneCycle LR schedule (Smith & Topin, 2017).

### A.3 Linear probe training

A linear classifier is applied to the embeddings generated by all the pretrained models for species-level classification. The parameters of the model are learned using stochastic gradient descent with a constant learning rate of 0.01, momentum $\mu = 0.9$ and weight $\lambda = 1 \times 10^{-5}$.

For the hyperparameter search, shown in Table 6, our linear probe is performed using the same methodology as the fine-tuning stage, except the encoder parameters are frozen.

Table 6: Search over the space of $k$-mer tokenization length and transformer architectures (number of layers and heads). For fine-tuned and linear probe, we show the class-balanced accuracy (%) on the closed-world `val` partition, and for 1-NN probe, we show the class-balanced accuracy on the `val_unseen` partition. Bold: architecture with highest accuracy for the row. Underlined: second highest accuracy.

| Evaluation | 4 layers, 4 heads | | | | 6 layers, 6 heads | | | | 12 layers, 12 heads | | | |
| --- | --- | --- | --- | --- | --- | --- | --- | --- | --- | --- | --- | --- |
| | $k=2$ | $k=4$ | $k=6$ | $k=8$ | $k=2$ | $k=4$ | $k=6$ | $k=8$ | $k=2$ | $k=4$ | $k=6$ | $k=8$ |
| Fine-tuned | 93.8 | 97.8 | 98.7 | **98.9** | 92.4 | 97.9 | 49.4 | 98.7 | 93.8 | 98.1 | 0.0 | 0.0 |
| Linear probe | 32.2 | 79.8 | 76.4 | **97.1** | 34.3 | 58.9 | 8.9 | 79.7 | 16.4 | 3.2 | 0.0 | 0.0 |
| 1-NN | 43.1 | **50.7** | 35.0 | 46.4 | 46.2 | 37.2 | 23.4 | 37.9 | 29.1 | 28.3 | 0.0 | 0.1 |

## B  Zero-Shot Clustering — Additional Experiments

As described in §4.3, we performed a series of zero-shot clustering experiments to establish how pretrained image and DNA models could handle the challenge of grouping together repeat observations of novel/unseen species. Our methodology is illustrated in Figure 6.

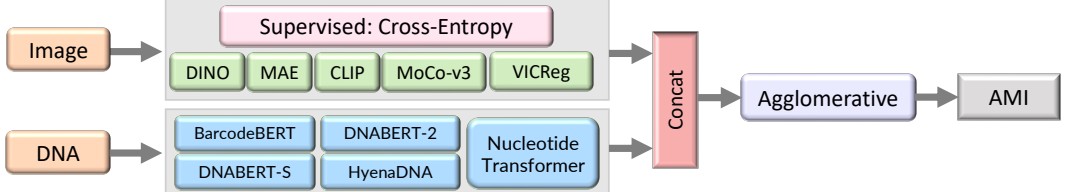

Figure 6: **Zero-shot clustering methodology.** Images and DNA are each passed through one of several pretrained encoders. These representations are clustered with Agglomerative Clustering.

### B.1 Experiment resources

All zero-shot clustering experiments were performed on a compute cluster with the job utilizing two CPU cores (2x Intel Xeon Gold 6148 CPU @ 2.40GHz) and no more than 20 GB of RAM. The typical runtime per experiment was around 4.5 hours.

## B.2 Accounting for Duplicated DNA Barcode Sequences

In our main experiments, we found that the performance of DNA-embedding clusterings greatly outperformed that of image-embeddings. However, it is worth considering that there are fewer unique DNA barcodes than images. The mean number of samples per barcode is around two. This provides clustering methods using DNA with an immediate advantage as some stimuli compare as equal and are trivially grouped together, irrespective of the encoder.

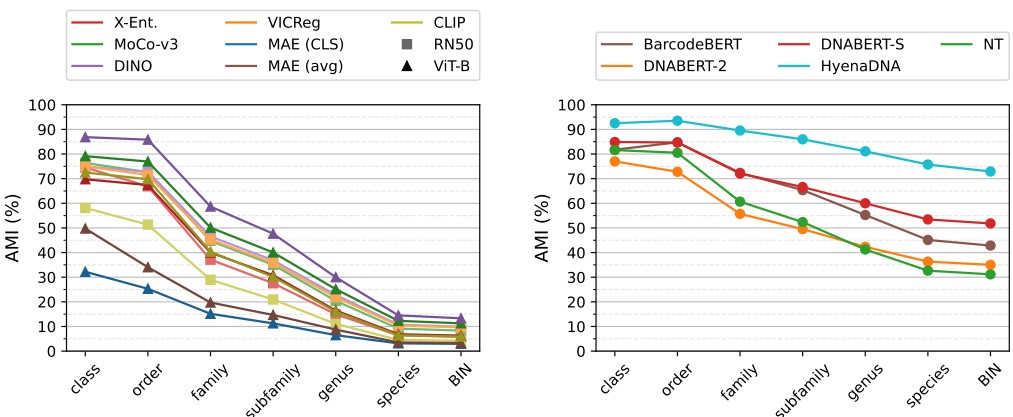

Figure 7: **Zero-shot clustering** AMI (%) performance across taxonomic ranks on `test` and `test_unseen` data, with **one sample per barcode**.

To account for this, we repeated our analysis with only one sample per barcode. Our results, shown in Figure 7, indicate that both image- and DNA-based clusterings are reduced in performance when the number of samples per barcode is reduced to one. This is explained in part by the fact that many species will be reduced to a single observation, which is challenging for clusterers to handle. We found that the performance of most DNA encoders fell by more than the image encoders when the number of samples per barcode was reduced to one. However, the DNA embeddings still produce clusterings in better agreement with the taxonomic labels than the image embeddings. In particular, the best-performing DNA encoder, HyenaDNA, still attained 75% agreement with the ground truth labels at the species-level clustering.

## B.3 Cross-modal embedding clustering

We additionally considered the effect of clustering the embeddings from both modalities at once, achieved by concatenating an image embedding and a DNA embedding to create a longer feature vector per sample. As shown in Table 7, we find that combining image features with DNA features results in a worse performance at species-level clustering.

In preliminary experiments (not shown) we found that the magnitude of the vectors greatly impacted the performance, as large image embeddings would dominate DNA embeddings with a smaller magnitude. We considered standardizing the embeddings before concatenation with several methods (L2-norm, element-wise z-score, average z-score) and found element-wise z-score gave the best performance, a step which we include in these results. Even with this, the performance falls when we add image embeddings to the DNA embeddings. We note that the best DNA-only encoder, HyenaDNA, has the largest drop in performance, which we hypothesize is because it has the shortest embedding dimensions of 128-d compared with NT (512-d) and the BERT-based models (768-d).

Table 7: **Cross-modal zero-shot clustering** AMI (%) performance, on `test` and `test_unseen` data, with **one sample per barcode**.

| Architecture | Image encoder | *Image-only* | BarcodeBERT | DNABERT-2 | DNABERT-S | HyenaDNA | NT |
|---|---|---|---|---|---|---|---|
| | | | | DNA encoder | | | |
| — | *DNA-only* | – | 47 | 52 | 63 | **81** | 36 |
| ResNet-50 | X-Ent. | 5 | 30 | 26 | 32 | 9 | 12 |
| | MoCo-v3 | 8 | 29 | 23 | 27 | 11 | 11 |
| | DINO | 11 | 31 | 28 | 31 | 15 | 14 |
| | VICReg | 10 | 30 | 26 | 30 | 13 | 13 |
| | CLIP | 6 | 25 | 21 | 25 | 9 | 9 |
| ViT-B | X-Ent. | 7 | 33 | 35 | 42 | 13 | 14 |
| | MoCo-v3 | 13 | 38 | 43 | 49 | 21 | 20 |
| | DINO | **15** | 38 | 45 | 51 | 23 | 21 |
| | MAE (CLS) | 5 | 33 | 33 | 40 | 10 | 13 |
| | MAE (avg) | 3 | 29 | 26 | 32 | 7 | 9 |
| | CLIP | 7 | 34 | 37 | 44 | 14 | 16 |

## C  Multi-Modal Learning — Additional Experiments

As described in §4.4, we trained a multimodal model with an aligned embedding space across the images, DNA, and taxonomic labels. Our methodology is illustrated in Figure 8.

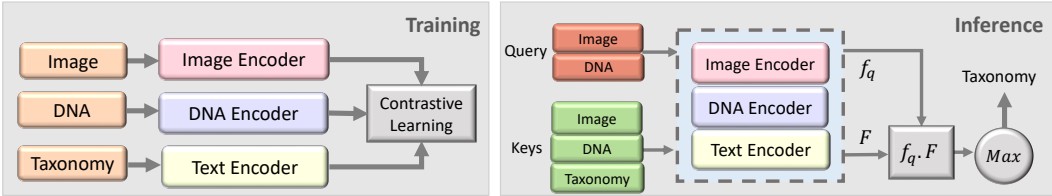

Figure 8: **Multi-modal learning methodology.** Our experiments using CLIBD (Gong et al., 2024) are conducted in two steps. (1) Training: Multiple modalities, including RGB images, textual taxonomy, and DNA sequences, are encoded separately, and trained using a contrastive loss function. (2) Inference: Image vs DNA embedding is used as a query, and compared to the embeddings obtained from a database of image, DNA and text (keys). The cosine similarity is used to find the closest key embedding, and the corresponding taxonomic label is used to classify the query.

### C.1  Model training and inference

We illustrate our model training and inference methodology in Figure 8. For our multimodal model, we start with pretrained encoders for image, DNA, and taxonomic labels. We use contrastive learning to fine-tune the image, DNA, and text encoders. During inference, we compare the embedding of the query image or DNA input to a key database of embeddings from images, DNA, or taxonomy labels using cosine similarity, and we predict the query's taxonomy based on the taxonomy of the closest retrieved key embeddings.

### C.2  Additional experiments

In the main paper, we reported the macro accuracy of our models. In Table 8, we report the micro accuracy to compare performance when averaged over individual samples rather than classes. The results show similar trends to the macro accuracy (Figure 8), with the model trained on the

Table 8: Top-1 *micro* accuracy (%) on the test set for using different amount of training data (1 million vs 5 million records from BIOSCAN-5M) and different combinations of aligned embeddings (image, DNA, text) during contrastive training. We show results for using image-to-image, DNA-to-DNA, and image-to-DNA query and key combinations. As a baseline, we show the results prior to contrastive learning (no alignment). We report the accuracy for seen and unseen species, and the harmonic mean (H.M.) between these (bold: highest acc.).

| Taxon | # Records | Aligned embeddings | | | DNA-to-DNA | | | Image-to-Image | | | Image-to-DNA | | |
|---|---|---|---|---|---|---|---|---|---|---|---|---|---|
| | | Img | DNA | Txt | Seen | Unseen | H.M. | Seen | Unseen | H.M. | Seen | Unseen | H.M. |
| Order | — | ✗ | ✗ | ✗ | 98.9 | 99.3 | 99.1 | 94.2 | 97.0 | 95.6 | 18.3 | 14.7 | 16.3 |
| | 1M | ✓ | ✓ | ✓ | **100.0** | **100.0** | **100.0** | 99.3 | 99.6 | 99.5 | 98.7 | 99.2 | 98.9 |
| | 5M | ✓ | ✓ | ✗ | **100.0** | **100.0** | **100.0** | **99.5** | **99.7** | **99.6** | **99.4** | 99.5 | **99.5** |
| | 5M | ✓ | ✓ | ✓ | **100.0** | **100.0** | **100.0** | **99.5** | **99.7** | **99.6** | 99.3 | **99.6** | **99.5** |
| Family | — | ✗ | ✗ | ✗ | 96.5 | 97.3 | 96.9 | 72.9 | 76.0 | 74.4 | 1.7 | 1.9 | 1.8 |
| | 1M | ✓ | ✓ | ✓ | 99.8 | 99.8 | 99.8 | 95.5 | 96.8 | 96.2 | 90.6 | 89.1 | 89.9 |
| | 5M | ✓ | ✓ | ✗ | **99.9** | **100.0** | 99.9 | 96.8 | 97.9 | 97.4 | 94.0 | 93.1 | 93.5 |
| | 5M | ✓ | ✓ | ✓ | **99.9** | **100.0** | **100.0** | **97.0** | **98.3** | **97.7** | **94.6** | **94.4** | **94.5** |
| Genus | — | ✗ | ✗ | ✗ | 94.0 | 93.5 | 93.7 | 47.8 | 47.0 | 47.4 | 0.2 | 0.0 | 0.1 |
| | 1M | ✓ | ✓ | ✓ | 99.3 | 98.8 | 99.0 | 86.0 | 85.9 | 86.0 | 68.1 | 52.3 | 59.2 |
| | 5M | ✓ | ✓ | ✗ | **99.6** | **99.8** | **99.7** | 90.6 | 91.6 | 91.1 | **79.5** | 65.0 | 71.5 |
| | 5M | ✓ | ✓ | ✓ | **99.6** | **99.8** | **99.7** | **91.0** | **92.1** | **91.5** | 79.3 | **66.3** | **72.2** |
| Species | — | ✗ | ✗ | ✗ | 91.6 | 84.8 | 88.1 | 31.9 | 19.1 | 23.9 | 0.0 | 0.0 | 0.0 |
| | 1M | ✓ | ✓ | ✓ | 98.3 | 95.0 | 96.6 | 75.1 | 57.5 | 65.1 | 47.9 | 10.4 | 17.0 |
| | 5M | ✓ | ✓ | ✗ | **98.9** | 97.4 | 98.2 | 82.7 | **68.3** | **74.8** | **64.2** | **18.7** | **29.0** |
| | 5M | ✓ | ✓ | ✓ | **98.9** | 97.7 | 98.3 | 82.8 | 67.6 | 74.4 | 61.7 | 17.8 | 27.7 |

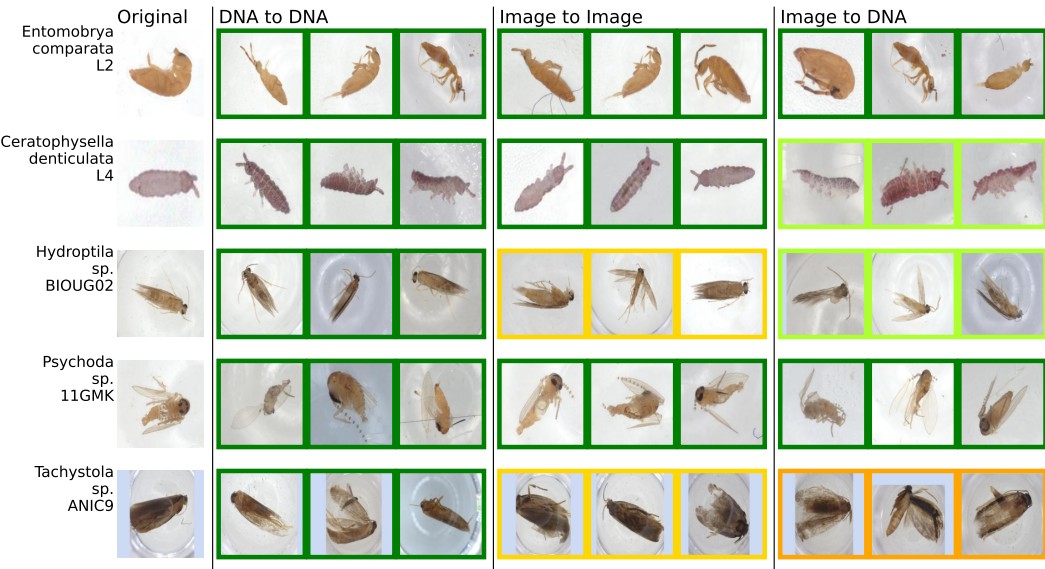

Figure 9: *Example query-key pairs.* Top-3 nearest specimens from the unseen validation-key dataset retrieved based on the cosine-similarity for DNA-to-DNA, image-to-image, and image-to-DNA retrieval. Box colour indicates whether the retrieved samples had the same species (green), genus (light-green), family (yellow), or order (orange) as the query.

BIOSCAN-5M dataset performing best for broader taxa, especially in image-to-image and image-to-DNA inference setups. Results are more mixed at the species level due in part to the challenge of species classification, highlighting the importance of further research at this fine-grained level.

## C.3 Retrieval examples

Figure 9 shows image retrieval examples using images as queries and DNA as keys. These demonstrate the ability of the model to classify taxonomy based on retrieval and the visual similarities of the retrieved images corresponding to the most closely matched DNA embeddings.

