# BIOSCAN-5M: A Multimodal Dataset for Insect Biodiversity Supplementary Materials

**Zahra Gharaee**[3*]**, Scott C. Lowe**[5*]**, ZeMing Gong**[4*]**, Pablo Millan Arias**[3*]**,**
**Nicholas Pellegrino**[3]**, Austin T. Wang**[4]**, Joakim Bruslund Haurum**[7]**,**
**Iuliia Zarubiieva**[2,5]**, Lila Kari**[3]**,**
**Dirk Steinke**[1,2†]**, Graham W. Taylor**[2,5†]**, Paul Fieguth**[3†]**, Angel X. Chang**[4,6†]

[1]Centre for Biodiversity Genomics, [2]University of Guelph, [3]University of Waterloo,
[4]Simon Fraser University, [5]Vector Institute, [6]Alberta Machine Intelligence Institute (Amii),
[7]Aalborg University and Pioneer Centre for AI
https://biodiversitygenomics.net/5M-insects/

## S1 Dataset

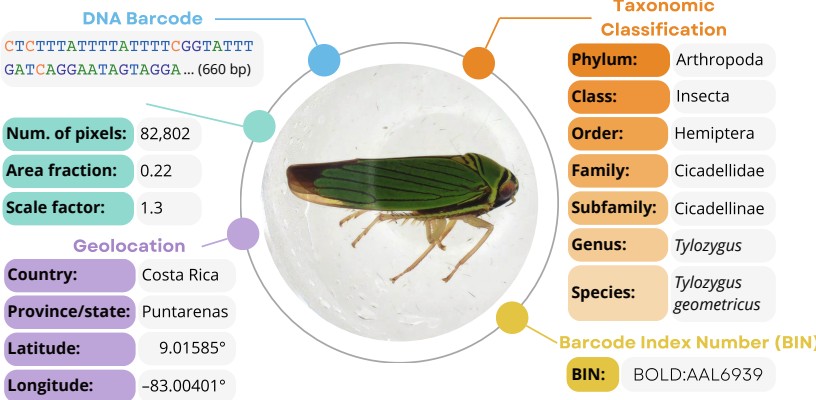

Figure S1: The BIOSCAN-5M Dataset provides taxonomic labels, a DNA barcode sequence, barcode index number, a high-resolution image along with its cropped and resized versions, as well as size and geographic information for each sample.

## S2 Ethics and responsible use

The BIOSCAN project was instigated by the International Barcode of Life (iBOL) Consortium, which has collected a large dataset of manually-labelled images of organisms (International Barcode of Life Consortium, 2024; Steinke et al., 2024). As part of our project, we conducted a thorough review to identify any potential ethical issues related to the inclusion of our data sources. After careful evaluation, we did not find any ethical concerns. Therefore, we confirm that this work adheres to all relevant ethical standards and guidelines.

---

[*]Joint first author. [†]Joint senior/last author.

38th Conference on Neural Information Processing Systems (NeurIPS 2024) Track on Datasets and Benchmarks.

## S3 Dataset availability and maintenance

To explore more about the BIOSCAN-5M dataset, kindly visit the following landing page: https://biodiversitygenomics.net/5M-insects/.

The BIOSCAN-5M dataset and all its contents are available in a GoogleDrive Folder. The Google Drive folder serves as the primary repository for the BIOSCAN-5M dataset, ensuring ongoing maintenance and the potential addition of new content as necessary. It will be gradually updated to address any data issues that may arise.

The Google Drive folder contains the following dataset contents:

- **BIOSCAN_5M_IMAGES**: This directory contains images:
  - `BIOSCAN_5M_original_full`: The original full-size images.
  - `BIOSCAN_5M_original_256`: The original images resized to 256 pixels on their shorter side.
  - `BIOSCAN_5M_cropped`: The cropped images.
  - `BIOSCAN_5M_cropped_256`: The cropped images resized to 256 pixels on their shorter side.
- **BIOSCAN_5M_METADATA**: This directory contains metadata:
  - `BIOSCAN_5M_Insect_Dataset_metadata_MultiTypes.zip`: A zip file containing both CSV and JSON formats of the metadata file.
- **BIOSCAN_5M_CropTool**: This directory contains our cropping tool components:
  - `bounding_box/BIOSCAN_5M_Insect_bbox.tsv`: A TSV file that includes bounding box information obtained from our cropping tool.
  - `checkpoint/BIOSCAN_5M_Insect_cropping_tool.ckpt`: The model checkpoint used to crop the original full-size images, which generated the cropped images of the BIOSCAN-5M dataset.

Additionally, the dataset is released on several platforms, including Zenodo, Kaggle, and HuggingFace.

We provide a code repository for dataset manipulation, which supports tasks like reading images and metadata, cropping images, statistical processing, dataset splitting into pretrain, train, and evaluation, as well as running benchmark experiments presented in the BIOSCAN-5M paper. To access the BIOSCAN-5M code repository, please visit https://github.com/bioscan-ml/BIOSCAN-5M.

Additionally, we provide a Python package for working with the BIOSCAN-5M dataset, designed in the style of torchvision's `VisionDataset` class, which can be installed with `pip install bioscan-dataset`. For usage details, please visit https://bioscan-dataset.readthedocs.io/.

## S4 Licensing

Table S1 shows all the copyright associations related to the BIOSCAN-5M dataset with the corresponding names and contact information.

Table S1: Copyright associations related to the BIOSCAN-5M dataset

| Copyright Associations | Name & Contact |
| --- | --- |
| Image Photographer | CBG Robotic Imager |
| Copyright Holder | CBG Photography Group |
| Copyright Institution | Centre for Biodiversity Genomics (email: CBGImaging@gmail.com) |
| Copyright License | Creative Commons Attribution 3.0 Unported (CC BY 3.0) |
| Copyright Contact | collectionsBIO@gmail.com |
| Copyright Year | 2021 |

The authors state that they bear all responsibility in case of violation of usage rights.

## S5 RGB images

The BIOSCAN-5M dataset comprises resized and cropped images, as introduced in BIOSCAN-1M Insect (Gharaee et al., 2023). We have provided various packages of the BIOSCAN-5M dataset, detailed in Table S2, each tailored for specific purposes.

- original_full: The raw images of the dataset, typically 1024×768 pixels.
- cropped: Images after cropping with our cropping tool (see §S14.1).
- original_256: Original images resized to 256 on their shorter side (most 341×256 pixels).
- cropped_256: Cropped images resized to 256 on their shorter side.

Among these, the original_256 and cropped_256 packages are specifically provided for experimentation as they are small and easy to work with. Therefore, using our predefined split partitions, we provide per-split experimental packages in addition to the packages with all the original_256 and cropped_256 images.

Table S2: Various downloadable packages of the images comprising the BIOSCAN-5M dataset.

| Image set | Package | Partition(s) | Size (GB) | # Parts |
|---|---|---|---|---|
| original_full | BIOSCAN_5M_original_full.zip | All | 200 | 5 |
| cropped | BIOSCAN_5M_cropped.zip | All | 77.2 | 2 |
| original_256 | BIOSCAN_5M_original_256.zip | All | 35.2 | 1 |
| | BIOSCAN_5M_original_256_pretrain.zip | Pretrain | 31.7 | 1 |
| | BIOSCAN_5M_original_256_train.zip | Train | 2.1 | 1 |
| | BIOSCAN_5M_original_256_eval.zip | Evaluation | 1.4 | 1 |
| cropped_256 | BIOSCAN_5M_cropped_256.zip | All | 36.4 | 1 |
| | BIOSCAN_5M_cropped_256_pretrain.zip | Pretrain | 33.0 | 1 |
| | BIOSCAN_5M_cropped_256_train.zip | Train | 2.1 | 1 |
| | BIOSCAN_5M_cropped_256_eval.zip | Evaluation | 1.4 | 1 |

Accessing the dataset images is facilitated by the following directory structure used to organize the dataset images:

```
bioscan5m/images/[imgtype]/[split]/[chunk]/[processid.jpg]
```

where [imgtype] can be original_full, cropped, original_256, or cropped_256. The [split] values can be pretrain, train, val, test, val_unseen, test_unseen, key_unseen, or other_heldout. Note that the val, test, val_unseen, test_unseen, key_unseen, and other_heldout splits are within the evaluation partition of the original_256 and cropped_256 image packages.

The [chunk] is determined by using the first one or two characters of the MD5 checksum (in hexadecimal) of the processid. This method ensures that the chunk name is purely deterministic and can be computed directly from the processid. As a result, the pretrain split organizes files into 256 directories by using the first two letters of the MD5 checksum of the processid. For the train and other_heldout splits, files are organized into 16 directories using the first letter of the MD5 checksum. The remaining splits do not use chunk directories since each split has less than 50 k images.

## S6 Metadata file

To enrich the metadata of our published dataset, we provide integrated structured metadata conforming to Web standards. Our dataset's metadata file is titled **BIOSCAN_5M_Insect_Dataset_metadata**. We provide two versions of this file: one in CSV format (**.csv**) and the other in JSON-LD format (**.jsonld**). Accessing the dataset metadata files is facilitated by the following directory structure used to organize the dataset images:

```
bioscan5m/metadata/[type]/BIOSCAN_5M_Insect_Dataset_metadata.[type_extension]
```

In this structure, `[type]` refers to the file type of the metadata file, which can be either CSV or JSON-LD. The `[type_extension]` indicates the corresponding file extensions, which are `csv` for CSV files and `jsonld` for JSON-LD files.

Table S3 outlines the fields of the metadata file and the description of their contents.

Table S3: Table presents fields of the metadata file of BIOSCAN-5M dataset.

| | Field | Description | Type |
|---|---|---|---|
| 1 | processid | A unique number assigned by BOLD (International Barcode of Life Consortium). | String |
| 2 | sampleid | A unique identifier given by the collector. | String |
| 3 | taxon | Bio.info: Most specific taxonomy rank. | String |
| 4 | phylum | Bio.info: Taxonomic classification label at phylum rank. | String |
| 5 | class | Bio.info: Taxonomic classification label at class rank. | String |
| 6 | order | Bio.info: Taxonomic classification label at order rank. | String |
| 7 | family | Bio.info: Taxonomic classification label at family rank. | String |
| 8 | subfamily | Bio.info: Taxonomic classification label at subfamily rank. | String |
| 9 | genus | Bio.info: Taxonomic classification label at genus rank. | String |
| 10 | species | Bio.info: Taxonomic classification label at species rank. | String |
| 11 | dna_bin | Bio.info: Barcode Index Number (BIN). | String |
| 12 | dna_barcode | Bio.info: Nucleotide barcode sequence. | String |
| 13 | country | Geo.info: Country associated with the site of collection. | String |
| 14 | province_state | Geo.info: Province/state associated with the site of collection. | String |
| 15 | coord-lat | Geo.info: Latitude (WGS 84; decimal degrees) of the collection site. | Float |
| 16 | coord-lon | Geo.info: Longitude (WGS 84; decimal degrees) of the collection site. | Float |
| 17 | image_measurement_value | Size.info: Number of pixels occupied by the organism. | Integer |
| 18 | area_fraction | Size.info: Fraction of the original image the cropped image comprises. | Float |
| 19 | scale_factor | Size.info: Ratio of the cropped image to the cropped_256 image. | Float |
| 20 | inferred_ranks | An integer indicating at which taxonomic ranks the label is inferred. | Integer |
| 21 | split | Split set (partition) the sample belongs to. | String |
| 22 | index_bioscan_1M_insect | An index to locate organism in BIOSCAN-1M Insect metadata. | Integer |
| 23 | chunk | The packaging subdirectory name (or empty string) for this image. | String |

## S7   Comparison between BIOSCAN-5M and BIOSCAN-1M

The six key differences between BIOSCAN-1M and BIOSCAN-5M are as follows:

1. **Increased data volume:** BIOSCAN-5M contains five times as many samples as BIOSCAN-1M.

2. **Greater data diversity:** BIOSCAN-5M is collected from a broader range of geographic locations (3 countries in BIOSCAN-1M; 47 countries in BIOSCAN-5M) and encompasses a wider variety of insect life (1 class and 16 orders in BIOSCAN-1M; 10 classes and 55 orders in BIOSCAN-5M).

3. **Enhanced post-processing:** The taxonomic labels in BIOSCAN-5M underwent a rigorous data cleaning pipeline to identify and resolve inconsistencies in the original data, resulting in more reliable labels compared to those in BIOSCAN-1M.

4. **Geographic and specimen size data:** This information is available in BIOSCAN-5M but not in BIOSCAN-1M.

5. **Comprehensive partitioning support:** BIOSCAN-5M offers robust support for both closed-world and open-world tasks, whereas BIOSCAN-1M only supports closed-world partitioning.

6. **Enhanced benchmarking experiments:** BIOSCAN-1M included a baseline with an image-only model evaluated at order and family ranks. In contrast, BIOSCAN-5M features three baselines that leverage the multimodal aspects of the dataset (including DNA barcode sequences, textual taxonomic labels, and RGB images), allowing for performance exploration in both closed- and open-world settings.

## S8 Focus and objectives

We have released dataset splits for closed-world and open-world settings, using labelled species data for evaluation and reserving unlabelled data for pretraining. Our splitting approach and configurations offer valuable resources to the ML community. BIOSCAN-5M experiments evaluate down to the species level. Additionally, we benchmark three distinct tasks to showcase BIOSCAN-5M's multimodal utility in real-world applications: fine-grained taxonomic classification with DNA sequences, classification using DNA, images, and taxonomic labels, and clustering of DNA and image embeddings.

### S8.1 Leveraging unlabelled and multimodal data for enhanced taxonomic classification

It's important to note that taxonomic classification from images presents greater challenges compared to DNA barcodes, as illustrated by our clustering experiments; thus, paired data can be valuable even when unlabelled. Additionally, data not labelled at the species level remains useful for pretraining, highlighting the crucial role of unlabelled data in model development. In BIOSCAN-5M, we employ BERT-style masked sequence modelling to pretrain and encode DNA sequences, complemented by contrastive learning to align image and DNA embeddings. This pretraining approach enhances the model's ability to generalize across various applications.

### S8.2 Lack of utilization of geographic and size information in models

In BIOSCAN-5M, we focus on biological (taxonomic labels) and genetic (DNA barcode sequences and BIN) data for fine-grained taxonomic classification, intentionally excluding geographic and size information from our experiments. Our rationale is that while geographic and size data can help rule out certain species (e.g., knowing a sample was collected in North America excludes species not found there, and knowing a sample's size eliminates species that do not grow that large), they alone do not provide sufficient information for accurate species classification. In contrast, image and genetic data are often sufficient for accurate species-level predictions.

We believe that models incorporating geographic and size data will need to do so alongside image and genetic data. Therefore, models using only image and genetic information serve as valuable baselines for future work that combines these data types. Given the complexities of integrating geographic and size data into our models, we prioritized establishing a broad range of image and genetic baselines in this study and plan to explore the incorporation of geographic and size data in future research. We anticipate that effective use of this additional information will enhance model performance and look forward to the community's advancements in this area.

## S9 Dataset features statistics

This section provides additional information regarding the dataset, including a detailed statistical analysis of its diverse multimodal data types and processing methods.

### S9.1 Geographical information

The detailed statistical analysis of the geographical locations where the organisms were collected is presented in Table S4. This table indicates the number of distinct regions represented by country, province or state, along with their corresponding latitude and longitude. Additionally, Table S4 provides the count of labelled versus unlabelled records, as well as the class imbalance ratio (IR) for each location group within the dataset.

The latitude and longitude coordinates indicate that the dataset comprises 1,650 distinct regions with unique geographical locations shown by Table S4. However, a significant portion of the organisms—approximately 73.36%—were collected from the top 70 most populated regions, which represent only 4.24% of the total regions identified by their coordinates.

Figure S2 shows the distinct countries where the organisms were collected on the world map. The majority of the organisms, over 62%, were collected from Costa Rica.

Table S4: The statistics for the columns indicating geographical locations where the specimens are collected.

| Geo locations | Categories | Labelled | Labelled (%) | Unlabelled | Unlabelled (%) | IR |
|---|---|---|---|---|---|---|
| `country` | 47 | 5,150,842 | 100.00 | 8 | 0.00 | 325,631.6 |
| `province_state` | 102 | 5,058,718 | 98.21 | 92,132 | 1.79 | 1,243,427.0 |
| `coord-lat` | 1,394 | 5,149,019 | 99.96 | 1,831 | 0.04 | 556,352.0 |
| `coord-lon` | 1,489 | 5,149,019 | 99.96 | 1,831 | 0.04 | 618,931.0 |
| Location (lat, lon) | 1,650 | 5,149,019 | 99.96 | 1,831 | 0.04 | 520,792.0 |

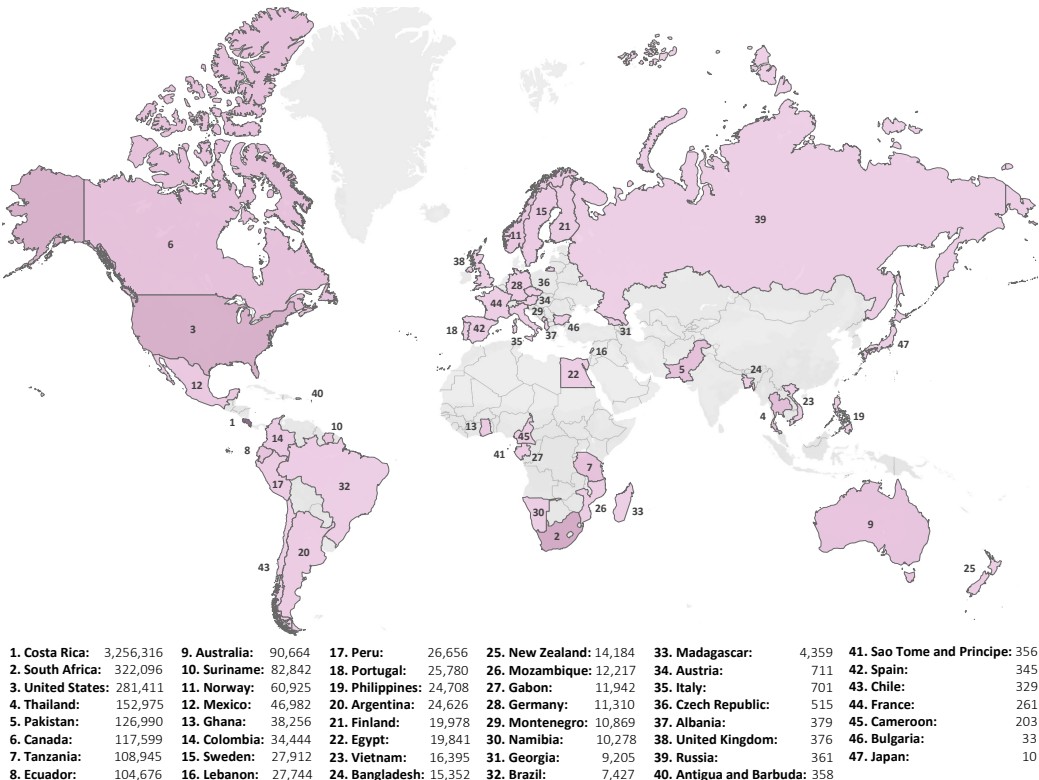

| | | | | | |
|---|---|---|---|---|---|
| **1. Costa Rica:** 3,256,316 | **9. Australia:** 90,664 | **17. Peru:** 26,656 | **25. New Zealand:** 14,184 | **33. Madagascar:** 4,359 | **41. Sao Tome and Principe:** 356 |
| **2. South Africa:** 322,096 | **10. Suriname:** 82,842 | **18. Portugal:** 25,780 | **26. Mozambique:** 12,217 | **34. Austria:** 711 | **42. Spain:** 345 |
| **3. United States:** 281,411 | **11. Norway:** 60,925 | **19. Philippines:** 24,708 | **27. Gabon:** 11,942 | **35. Italy:** 701 | **43. Chile:** 329 |
| **4. Thailand:** 152,975 | **12. Mexico:** 46,982 | **20. Argentina:** 24,626 | **28. Germany:** 11,310 | **36. Czech Republic:** 515 | **44. France:** 261 |
| **5. Pakistan:** 126,990 | **13. Ghana:** 38,256 | **21. Finland:** 19,978 | **29. Montenegro:** 10,869 | **37. Albania:** 379 | **45. Cameroon:** 203 |
| **6. Canada:** 117,599 | **14. Colombia:** 34,444 | **22. Egypt:** 19,841 | **30. Namibia:** 10,278 | **38. United Kingdom:** 376 | **46. Bulgaria:** 33 |
| **7. Tanzania:** 108,945 | **15. Sweden:** 27,912 | **23. Vietnam:** 16,395 | **31. Georgia:** 9,205 | **39. Russia:** 361 | **47. Japan:** 10 |
| **8. Ecuador:** 104,676 | **16. Lebanon:** 27,744 | **24. Bangladesh:** 15,352 | **32. Brazil:** 7,427 | **40. Antigua and Barbuda:** 358 | |

Figure S2: Global distribution of sample collection efforts. The countries are ranked by the number of samples collected.

## S9.2 Size information

Monitoring organism size is crucial as it can signal shifts in various factors affecting their lives, including food access, nutrition, and climate change (Sheridan & Bickford, 2011). For instance, in humans, limited access to nutrition correlates with a decrease in average height over generations (Steckel, 1995), reflecting environmental and economic changes. Tracking organism size offers insights into environmental shifts vital for biodiversity conservation (Hickling et al., 2006).

**Pixel count.** The raw dataset provides information about each organism's size by quantifying the total number of pixels occupied by the organism. This information is provided in the `image_measurement_value` field. Since the image capture settings are consistent for all images, irrespective of scale, as indicated by the organism's distance to the camera, the number of pixels occupied by the organism should approximate its size. Less than 1% of samples of the BIOSCAN-5M dataset do not have this information.

To provide a clearer understanding of the content in the `image_measurement_value` field, Figure S3 displays examples of original images along with their corresponding masks, highlighting the total

number of pixels occupied by an organism. To determine the real-world size of the organism based on the number of pixels, it is also important to have the pixel to metric scaling factor. For the original full sized images, most of the images are captured using a Keyence imaging system with a known pixel to millimetre scaling. See §S14.1 for details on the pixel scale and how to determine it for cropped and resized images.

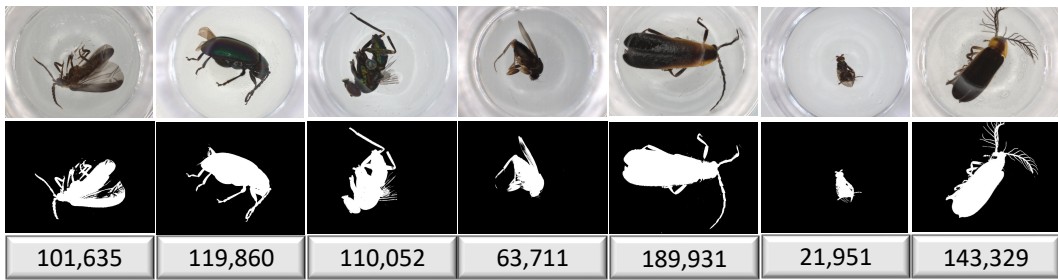

| 101,635 | 119,860 | 110,052 | 63,711 | 189,931 | 21,951 | 143,329 |

Figure S3: Examples of original images of the BIOSCAN-5M dataset, along with their respective total number of pixels (size) that occupy the image. The top row shows original images and the bottom row shows masks.

## S10  Dataset category distribution

Figure S4 illustrates the taxonomic class distribution within the rank order. For example, of the 99.9% of organisms labelled at the class level, approximately 71% are classified within the order *Diptera* of the class *Insecta*.

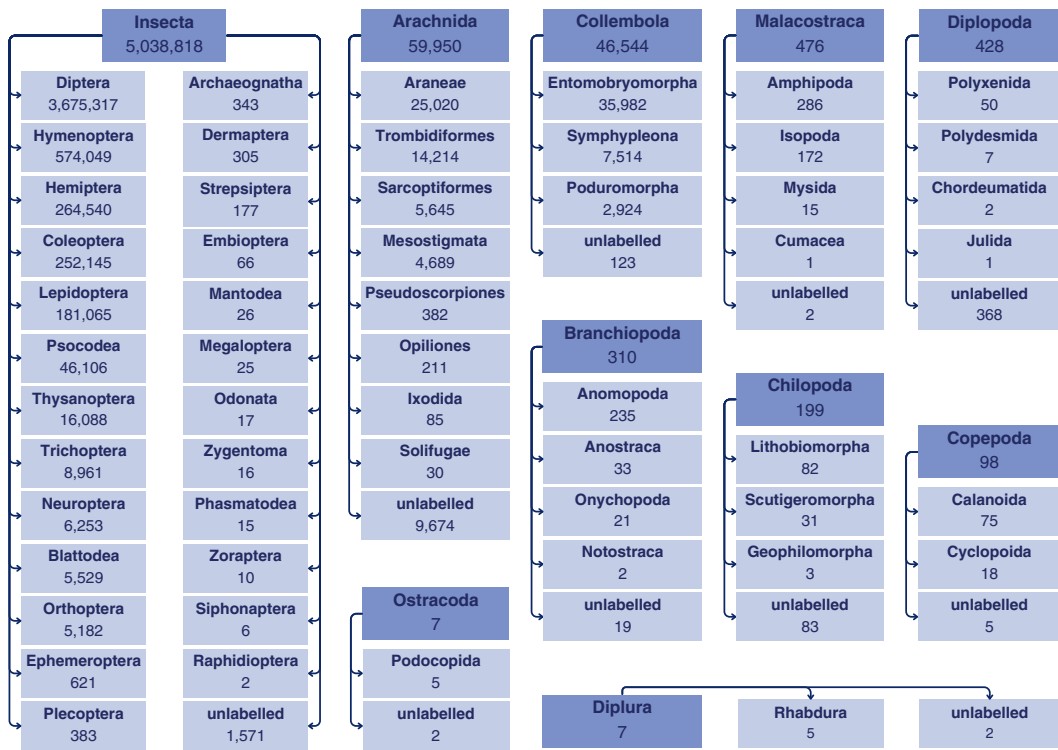

Figure S4: Distribution of taxonomic ranks in the BIOSCAN-5M dataset. Each darker cell represents a taxonomic class, while the lighter cells within each class represent the corresponding taxonomic orders. The numbers indicate the records belonging to each class and order. The *unlabeled* category denotes records assigned to a class but not to any specific order.

For detailed insights into the class distribution within the major categories of the BIOSCAN-5M dataset, Table S5 presents comprehensive statistics. This table provides the total number of categories across 7 taxonomic group levels and BINs, highlighting both the most and least densely populated ones. Additionally, it includes calculated means, medians, and standard deviations of the population vectors of all subcategories of each attribute.

Table S5: BIOSCAN-5M taxonomic and BIN categories distribution. For each attribute, we show the value which occurs most often in the dataset and the least populated value (in the event of a tie, we show an exemplar selected at random).

| Attributes | Categories | Most populated | | Least populated | | Mean | Median | Std. Dev. |
|---|---|---|---|---|---|---|---|---|
| | | Name | Size | Name | Size | | | |
| phylum | 1 | Arthropoda | 5,150,850 | Arthropoda | 5,150,850.0 | | | |
| class | 10 | Insecta | 5,038,818 | Ostracoda | 7 | 514,683.7 | 369.0 | 1,508,192.8 |
| order | 55 | Diptera | 3,675,317 | Cumacea | 1 | 93,363.4 | 172.0 | 495,969.5 |
| family | 934 | Cecidomyiidae | 938,928 | Pyrgodesmidae | 1 | 5,281.3 | 63.5 | 45,321.1 |
| subfamily | 1,542 | Metopininae | 323,146 | Bombyliinae | 1 | 953.7 | 23.0 | 9,092.8 |
| genus | 7,605 | Megaselia | 200,268 | chalMalaise9590 | 1 | 161.3 | 6.0 | 2,492.2 |
| species | 22,622 | Psychoda sp. 11GMK | 7,694 | Microcephalops sp. China3 | 1 | 20.9 | 2.0 | 139.5 |
| dna_bin | 324,411 | BOLD:AEO1530 | 35,458 | BOLD:ADT1070 | 1 | 15.8 | 2.0 | 146.4 |

# S11    DNA barcode statistics

This section presents the DNA barcode statistics and analysis for the BIOSCAN-5M dataset. We provide several different statistics to show how the diversity of DNA barcodes varies across the different taxonomic levels. In Table S6, we report the number of distinct barcodes, the Shannon diversity index (e.g. entropy), and the average pairwise distances between barcodes at different taxonomic ranks. The different analysis all show that at higher levels of taxa, there are more distinct barcodes, and that at the genus and species level, the lexical distance between different barcodes are much smaller than at the higher levels of taxa. Below we provide more details on how these statistics are computed.

## S11.1    Identical DNA barcodes: Statistical insights from the BIOSCAN-5M dataset

We compute and show in Table S6 the statistics for identical DNA barcode sequences across taxonomic ranks, including the total number of distinct barcodes per rank, as well as the average, median, and standard deviation of barcodes counts across subgroups within each rank.

Based on the statistics in Table S6, the total number of identical DNA barcode sequences within each subgroup of a specific taxonomic rank is lower than the total number of DNA sequences corresponding to the labelled samples in that subgroup. This indicates that some samples share identical DNA barcodes, possibly due to sequencing limitations. Since DNA barcodes are merely short snippets, they alone do not fully capture the unique genetic characteristics of individual samples.

## S11.2    Analyzing genetic diversity with the Shannon Index

**Shannon Diversity Index (SDI).** The Shannon Diversity Index (SDI) (Shannon, 1948), which measures the entropy within a group, is an effective metric for measuring genetic diversity as it considers both barcode richness (the number of distinct barcodes) and evenness (the distribution of samples among those barcodes). A high prevalence of identical barcodes leads to lower evenness and, consequently, a reduced SDI, indicating limited diversity and redundancy in genetic makeup.

Incorporating duplicated barcodes allows the SDI to capture the prevalence of specific barcodes within the subgroup. If certain barcodes are common across samples, the index may reflect a dominant genetic signature, resulting in a lower SDI and suggesting reduced diversity. Conversely, a greater presence of distinct barcodes with even distributions yields a higher SDI, indicating a more diverse subgroup structure. This dual focus on richness and evenness underscores the SDI's value in assessing genetic diversity and elucidating the genetic relationships within a subgroup.

Table S6: The DNA barcode statistics for various taxonomic ranks in the BIOSCAN-5M dataset. We indicate the total number of unique barcodes for the samples labelled to a given rank, and the mean, median, and standard deviation of the number of unique barcodes within the subgroupings at that rank. We also show the average across subgroups of the Shannon Diversity Index (SDI) for the DNA barcodes, measured in bits. We report the mean and standard deviation of pairwise DNA barcode sequence distances, aggregated across subgroups for each taxonomic rank.

| Attributes | Categories | Unique Barcodes | | | | | Pairwise Distance | |
| | | Total | Mean | Median | Std. Dev. | Avg SDI | Mean | Std. Dev. |
|---|---|---|---|---|---|---|---|---|
| phylum | 1 | 2,486,492 | | | | 19.78 | 158 | 42 |
| class | 10 | 2,482,891 | 248,289 | 177 | 725,237 | 8.56 | 166 | 103 |
| order | 55 | 2,474,855 | 44,997 | 57 | 225,098 | 7.05 | 128 | 53 |
| family | 934 | 2,321,301 | 2,485 | 46 | 19,701 | 5.42 | 90 | 46 |
| subfamily | 1,542 | 657,639 | 426 | 17 | 3,726 | 4.28 | 78 | 51 |
| genus | 7,605 | 531,109 | 70 | 5 | 1,061 | 2.63 | 50 | 39 |
| species | 22,622 | 202,260 | 9 | 2 | 37 | 1.46 | 17 | 18 |

We compute the Shannon Diversity Index (SDI) for each subgroup, $T$, within a taxonomic rank as

$$\text{SDI}_T = -\sum_{i=1}^{N} p_i \log_2(p_i),\tag{1}$$

where $N$ is the number of unique DNA barcodes within a subgroup, and $p_i$ is the fraction of samples in subgroup $T$ which have the $i$-th barcode.

In Table S6, we report the average SDI (Avg SDI) for each taxonomic rank by computing $\text{SDI}_T$ for each subgroup and then averaging these values across all subgroups within the respective rank. From the Table S6, the Avg SDI values indicate a high level of biodiversity at the phylum (19.78) and class (8.56) levels, suggesting a rich community with a wide variety of taxa. However, as we move down the taxonomic hierarchy, the index values decline significantly, reaching the lowest point at the species level (1.46). This pattern suggests that while there is a diverse range of phyla and classes, the distribution of species within these groups is uneven, indicating the presence of a few dominant species or genera.

### S11.3 Pairwise distance analysis of identical DNA barcodes

**Damerau-Levenshtein Distance.** The Damerau-Levenshtein distance (Damerau, 1964) is a string-edit distance metric that measures the minimum number of operations required to transform one string into another. It is an extension of the standard Levenshtein distance (Levenshtein, 1966), which counts the number of single-character edits needed for transformation. The key difference is that the Damerau-Levenshtein distance also accounts for transpositions, i.e., when two adjacent characters are swapped. In the context of our DNA barcoding, it measures how similar or different two DNA sequences are by counting how many single-character changes (insertions, deletions, substitutions, or transpositions) are needed to make one sequence identical to another.

We report the average Damerau-Levenshtein pairwise distance between unique DNA barcodes at different taxonomic ranks in Table S6. To compute the statistics for the pairwise distances, we take each subgroup at every taxonomic rank, and only consider subgroups with sufficient number of distinct barcodes. For a given subgroup, if the total number of unique DNA barcode sequences is fewer than 4, the subgroup is not considered. If the total exceeds 1,000, up to 1,000 sequences are randomly sampled; otherwise, all sequences are included.

To compute the distances between barcodes, the sampled DNA barcode sequences are first aligned using the MAFFT alignment technique (Katoh & Standley, 2013). Next, the pairwise distances between aligned DNA barcodes are computed using the Damerau-Levenshtein metric, with a total of $n \times \frac{(n-1)}{2}$ comparisons (where $n$ is the number of DNA barcodes). The mean and standard deviation of these distance values are then computed within each subgroup and subsequently aggregated using the mean function across subgroups at each taxonomic rank.

The statistics (Table S6, right columns) indicate that as we progress from higher to lower taxonomic ranks (e.g., from `phylum` and `class` to `genus` and `species`), both the mean and standard deviation of pairwise genetic distances decrease. This reduction indicates that genetic differences between organisms become smaller as we move down the taxonomic hierarchy, meaning organisms at lower ranks are more genetically similar to each other compared to those at higher ranks. For instance, `species` within the same `genus` tend to have much more similar DNA sequences than `families` within an `order` or `orders` within a `class`. This pattern aligns with the hierarchical structure of biological classification, where organisms are grouped based on increasing genetic relatedness as we move to finer taxonomic levels.

At the same time, the larger standard deviations observed at higher taxonomic ranks, such as `class` and `order`, reflect greater variability in genetic distances, suggesting a broader range of genetic diversity at these levels. Conversely, at lower ranks, such as `genus` and `species`, the smaller mean and standard deviation of pairwise distances highlight closer genetic relationships. However, these reduced distances can pose challenges for classification since the differences between closely related species become subtle.

This emphasizes the need for finer genetic markers or additional traits beyond pairwise distances to accurately distinguish between organisms, especially at the `species` level, where genetic distinctions can be minimal. Incorporating multimodal data, such as combining DNA sequences with images, can help address this challenge by providing complementary information. While DNA sequences offer insights into genetic differences, images capture morphological traits that may not be reflected in the genetic data. This multimodal approach can enhance classification accuracy, particularly when distinguishing between closely related species.

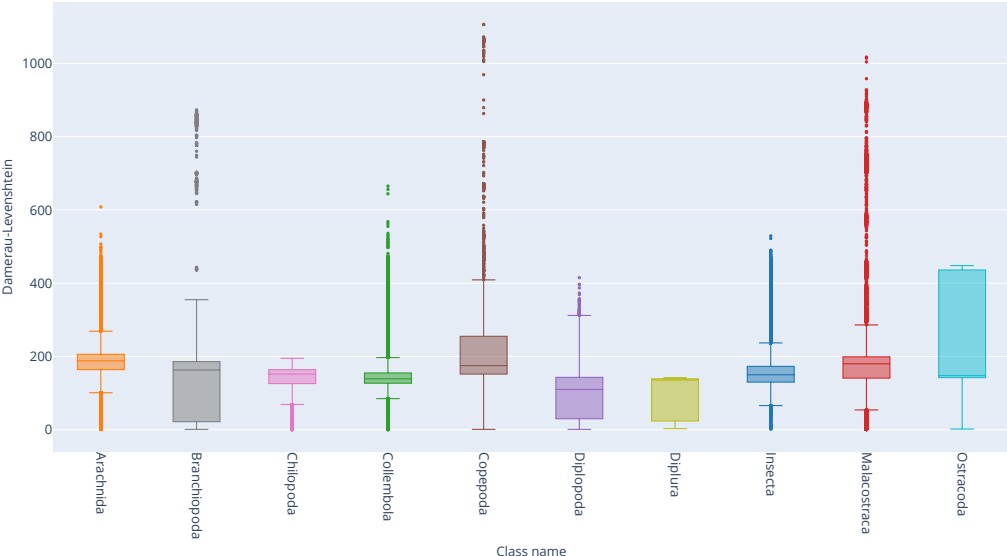

Figure S5: **Distribution of pairwise distances of subgroups of class.** The x-axis shows the subgroup categories sorted alphabetically.

Figure S5, Figure S6 and Figure S7 provide a visual representation of the statistics of pairwise distances computed in Table S6 for taxonomic ranks `class`, `order`, and `species`, respectively. The Interquartile Range (IQR) is a measure of statistical dispersion that describes the range within which the central 50% of the pairwise distances lies. It is calculated as the difference between the third quartile ($Q_3$) and the first quartile ($Q_1$) of the data,

$$\text{IQR} = Q_3 - Q_1,$$

where $Q_1$ is the 25th percentile of the data, and $Q_3$ is the 75th percentile. The line inside the box represents the median ($Q_2$) of the data. The height of the box illustrates the IQR. The lines extending from the box (whiskers) indicate the range of the data outside the IQR, typically extending up to 1.5 times the IQR from the quartiles, which help identify the spread of the data.

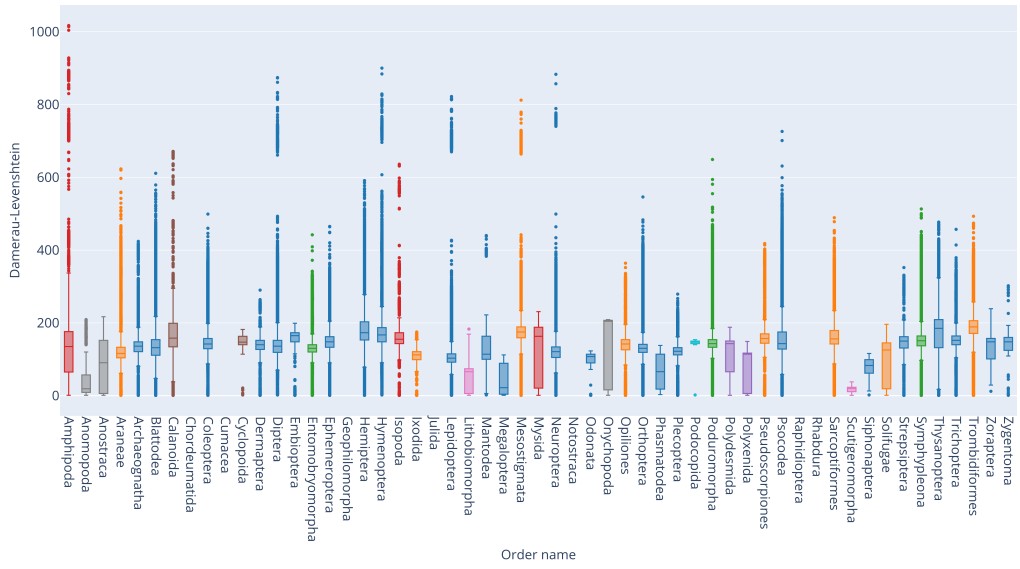

Figure S6: **Distribution of pairwise distances of subgroups of order.** The x-axis shows the subgroup categories sorted alphabetically.

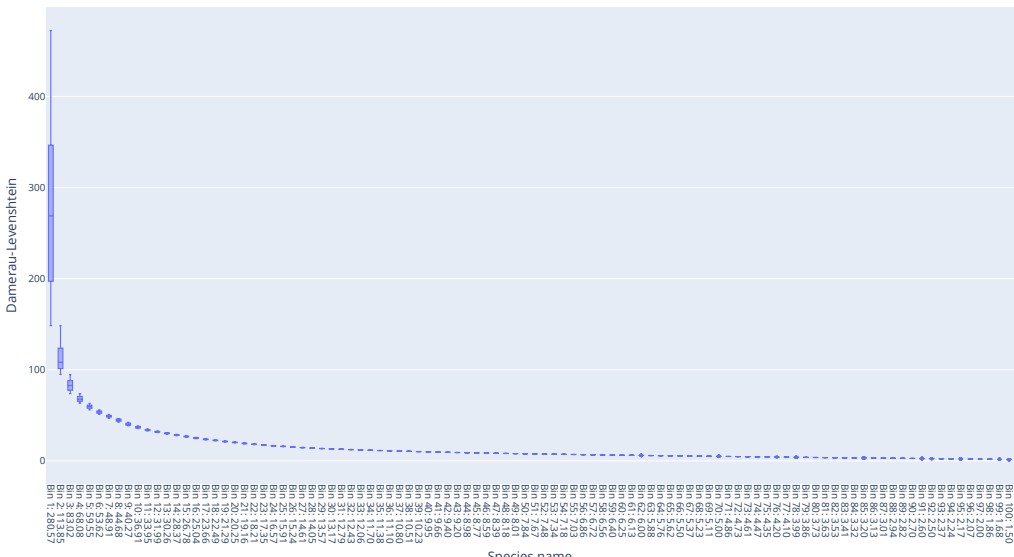

Figure S7: **Distribution of pairwise distances of subgroups of species.** Among the species, there are 8,372 distinct subgroups with sufficient identical barcodes for calculating pairwise distances, which makes visualization challenging. To address this, the groups are sorted in descending order based on their mean distances and partitioned into 100 bins. These bins are used to plot the distribution of pairwise distances within the species rank. The mean distance of each bin is displayed along the x-axis.

A small IQR (e.g., Collemboda in Figure S5) indicates that the pairwise distances among DNA barcode sequences within the group are tightly clustered around the median, suggesting that the sequences are similar to one another. This homogeneity may imply that the groups consist of closely related species or individuals with minimal genetic divergence, possibly due to a recent common ancestor.

Conversely, a large IQR (e.g., Ostracoda in Figure S5) signifies significant variability in the pairwise distances among sequences within a group, indicating a wider range of genetic diversity. This heterogeneity suggests that the groups may encompass genetically diverse species or populations with notable evolutionary divergence. Additionally, the presence of a large IQR may point to potential outliers—sequences that differ substantially from the majority—which could warrant further investigation to understand the underlying genetic variations.

If the whiskers are long while the IQR is small (e.g., Malacostraca in Figure S5), it implies that there are outlier values or a wider distribution of data points beyond the central cluster, highlighting the presence of variability in the dataset that may be worth investigating further.

If the median $Q_2$ is closer to $Q_1$ (e.g., Copepoda in Figure S5), the distribution is positively skewed, with most data points concentrated at the lower end and fewer but larger values at the higher end. Conversely, if the median is closer to $Q_3$ (e.g., Branchiopoda in Figure S5), the distribution is negatively skewed, with more values at the higher end and fewer, smaller values at the lower end.

Note that in all taxonomic ranks except for `species`, a random selection of 1,000 records is made for subgroups with more than 1,000 samples. For the `species` rank, all subgroups with a large number of records are included in the pairwise distance calculations. Some taxonomic ranks contain extremely large subgroups, such as *Arthropoda* in `phylum` and *Insecta* in `class`, each with over 2 million unique DNA records. Consequently, the 1,000 selected records may not fully represent the pairwise distances within the large subgroups. Due to computational limitations—since 1,000 records result in about 500 k unique pairwise distance computation—we adhere to this rule of selecting a random subset of 1,000 records.

## S12 Insect vs non-insect organisms

Focusing on *Insecta* as the most populous group at the `class` level, we present its detailed statistical records for DNA, BIN, and various taxonomic ranks in Table S7.

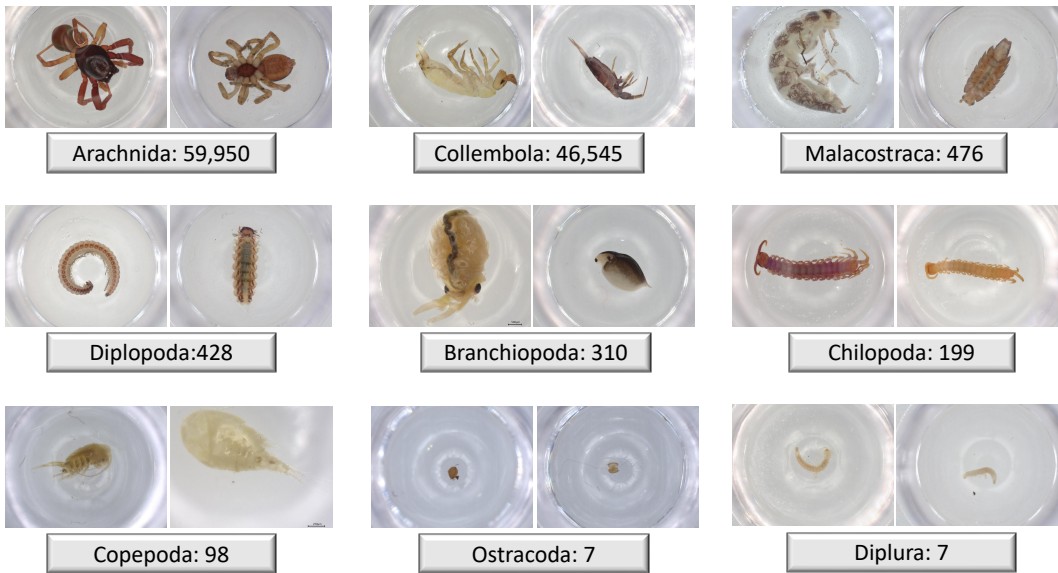

Figure S8: Examples of original images of non-insect organisms from the BIOSCAN-5M dataset. Below each image, the `class` name and its population within the BIOSCAN-5M dataset are displayed.

Figure S4 shows the class distribution within the taxonomic rank `class`, with 99.9% of organisms labelled at this level, of which 97.8% belong to the `class` *Insecta*. Figure S8 displays original images of non-insect taxonomic classes from the BIOSCAN-5M dataset, which includes a total of 137,479 organisms.

Table S7: Detailed statistical records for DNA, BIN and taxonomic ranks within `class` *Insecta* of the BIOSCAN-5M dataset.

| Attributes | Categories | Labelled | Labelled (%) | Unlabelled | Unlabelled (%) | IR |
|---|---|---|---|---|---|---|
| order | 25 | 5,037,247 | 99.97 | 1,571 | 0.03 | 1,837,658 |
| family | 681 | 4,853,383 | 96.32 | 185,435 | 3.68 | 938,928 |
| subfamily | 1,305 | 1,431,962 | 28.42 | 3,606,856 | 71.58 | 323,146 |
| genus | 6,897 | 1,188,043 | 23.58 | 3,850,775 | 76.42 | 200,268 |
| species | 21,512 | 450,215 | 8.93 | 4,588,603 | 91.07 | 7,694 |
| taxon | 26,603 | 5,038,818 | 100.00 | 0 | 0.00 | 925,520 |
| dna_bin | 311,743 | 5,025,921 | 99.74 | 12,897 | 0.26 | 35,458 |
| dna_barcode | 2,423,704 | 5,038,818 | 100.00 | 0 | 0.00 | 3,743 |

## S13 Limitations and challenges

### S13.1 Fine-grained classification

The BIOSCAN-5M dataset offers detailed biological features for each organism by annotating images with multi-grained taxonomic ranks. The class imbalance ratio (IR) across taxonomic groups reveals significant disparities in sample sizes between the majority class (with the most samples) and minority classes (with fewer samples). Notably, among the 55 distinct `orders`, *Diptera* accounts for approximately 71% of the total organisms. Figure S9 illustrates various `species` within the `order` *Diptera*, highlighting the high similarity among images of distinct categories, which poses additional challenges for downstream image classification tasks.

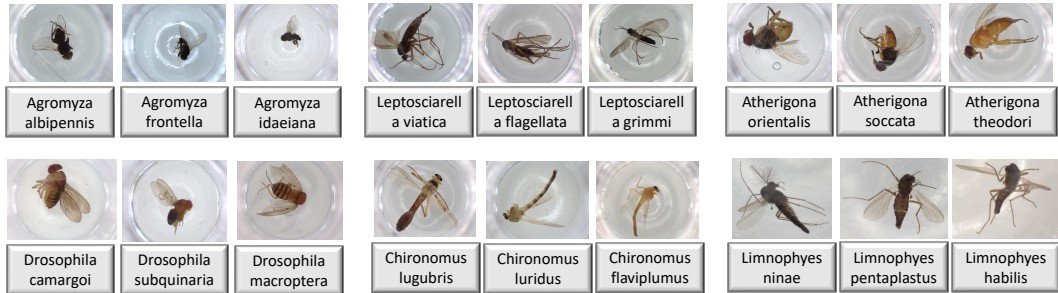

Figure S9: Sample images of distinct `species` from the `order` *Diptera*, which comprises about 71% of BIOSCAN-5M dataset. High similarity between samples of different `species` highlights significant image classification challenges.

### S13.2 Accessing ground-truth labels

The BIOSCAN-5M dataset exposes a limitation regarding labelling. The number of labelled records sharply declines as we delve deeper into taxonomic ranking groups, particularly when moving towards finer-grained taxonomic ranks beyond the `family` level. In fact, over 80% of the organisms lack taxonomic labels for ranks such as `subfamily`, `genus` and `species`. This circumstance poses a significant challenge for conducting taxonomic classification tasks. However, this limitation also opens doors to opportunities for research in various domains. The abundance of unlabelled data presents avenues for exploration in clustering, unsupervised, semi-supervised, and self-supervised learning paradigms, allowing for innovative approaches to data analysis and knowledge discovery.

### S13.3 Sampling Bias

The BIOSCAN-5M dataset also exposes a sampling bias as a result of the locations where and the methods through which organisms were collected, as depicted by Figure S2.

# S14 Data processing

To optimize our benchmark experiments using the BIOSCAN-5M dataset, we implemented two critical pre-processing steps on the raw dataset samples. These steps were necessary to enhance the efficiency and accuracy of our downstream tasks.

The first step involved image cropping and resizing. Due to the high resolution and large size of images in the dataset, processing the original images is both time-consuming and computationally expensive. Additionally, the area around the organism in each image is redundant for our feature extraction. To address these issues, we cropped the images to focus on the region of interest, specifically the area containing the organism. This step eliminated unnecessary background, reducing the data size and focusing the analysis on the relevant parts of the images. After cropping, we resized the images to a standardized resolution, further reducing the computational load and ensuring uniformity across all image samples.

The second step addressed inconsistencies in the taxonomic labels. In the raw dataset, we encountered identical DNA nucleotide sequences labelled differently at certain taxonomic levels, likely due to human error (e.g., typos) or disagreements in taxonomic naming conventions. Such discrepancies posed significant challenges for our classification tasks involving images and DNA barcodes. To address this, we implemented a multi-step cleaning process for the taxonomic labels. We identified and flagged inconsistent labels associated with identical DNA sequences and corrected typographical errors to ensure accurate and consistent naming.

We present additional details of our pre-processing steps in the following section.

## S14.1 Image processing details

The BIOSCAN-5M dataset contains resized and cropped images following the process in BIOSCAN-1M Insect (Gharaee et al., 2023). We resized images to 256 px on the smaller dimension. As in BIOSCAN-1M, we opt to conduct experiments on the cropped and resized images due to their smaller size, facilitating efficient data loading from disk.

**Cropping.** Following BIOSCAN-1M (Gharaee et al., 2023), we develop our cropping tool by fine-tuning a DETR (Carion et al., 2020) model with a ResNet-50 (He et al., 2016) backbone (pretrained on MSCOCO, Lin et al., 2014) on a small set of 2,837 insect images annotated using the Toronto Annotation Suite[1].

For BIOSCAN-1M, the DETR model was fine-tuned using 2,000 insect images (see Section 4.2 of Gharaee et al., 2023 for details). While the BIOSCAN-1M cropping tool worked well in general, there are some images for which the cropping was poor. Thus, we took the BIOSCAN-1M cropping tool checkpoint, and further fine-tuned the model for BIOSCAN-5M using the same 2,000 images and an additional 837 images that were not well-cropped previously. We followed the same training setup and hyperparameter settings as in BIOSCAN-1M and fine-tuned DETR on one RTX2080 Ti with batch size 8 and a learning rate of 0.0001.

Table S8: We compare the performance of the DETR model we used for cropping that was trained with the extra 837 images (NWC-837) that were previously not well-cropped to the model used for BIOSCAN-1M. We report the Average Precision (AP) and Average Recall (AR) computed on an additional validation set consisting of 100 images that were not-well cropped previously (NWC-100-VAL), as well as the images (IP-100-VAL + IW-150-VAL) used to evaluate the cropping tool's model used in BIOSCAN-1M. Our updated model performs considerably better on NWC-100-VAL, while given comparable performance on the original validation set of images.

| Dataset | Training data | NWC-100-VAL | | IP-100-VAL + IW-150-VAL | |
|---|---|---|---|---|---|
| | | AP[0.75] | AR[0.50:0.95] | AP[0.75] | AR[0.50:0.95] |
| BIOSCAN-1M | IP-1000 + IW-1000 | 0.257 | 0.485 | **0.922** | **0.894** |
| BIOSCAN-5M | IP-1000 + IW-1000 + NWC-837 | **0.477** | **0.583** | 0.890 | 0.886 |

---

[1] https://aidemos.cs.toronto.edu/annotation-suite/

Table S8 shows that our model with additional data achieves better cropping performance on an evaluation set of 100 images that were previously poorly cropped (NWC-100-VAL). Before cropping, we increase the size of the predicted bounding box by a fixed ratio $R = 1.4$ relative to the tight bounding box to capture some of the image background. If the bounding box extends beyond the image's edge, we pad the image with maximum-intensity pixels to align with the white background. These processes are the same as used by Gharaee et al. (2023). After cropping, we save the cropped-out bounding box.

**Resizing.** After cropping the image, we resize the image to 256 pixels on its smaller side while maintaining the aspect ratio $(r = \frac{w}{h})$. As nearly all original images are 1024×768 pixels, our resized images are (nearly all) 341×256 pixels.

**Area fraction.** The `area_fraction` field in the metadata file indicates the proportion of the original image represented by the cropped image. This factor is calculated using the bounding box information predicted by our cropping tool and serves as an indicator of the organism's size. Figure S10 displays the bounding boxes detected by our cropping model, which we used to crop images in the BIOSCAN-5M dataset. The area fraction factor is calculated as follows:

$$f_a = \frac{w_c \, h_c}{w \, h} \tag{2}$$

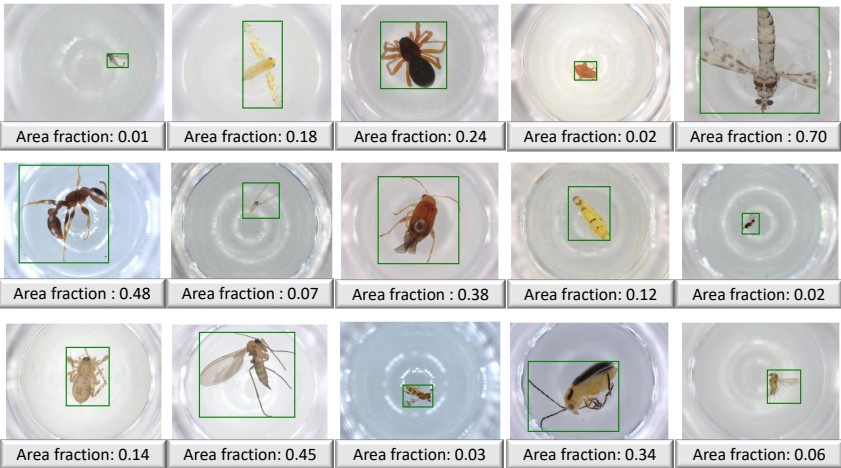

Figure S10: Examples of original images of organisms of the BIOSCAN-5M dataset with the bounding boxes detected by our cropping module. The area fraction value below each image shows how much of the original image is included in the crop.

**Scale factor.** When capturing images of physical objects, such as medical scans or biological samples, it is essential to ensure that measurements derived from these images accurately represent the real objects. To compute real-world sizes from captured images, a consistent relationship between pixel size and physical size is necessary. Therefore, we introduced the `scale_factor` field in the metadata file, which defines the ratio between the cropped image (`cropped`) and the cropped and resized image (`cropped_256`).

Assuming the original images $(I)$ have constant dimensions, width $(w)$ and height $(h)$, the cropped images $(I_c)$ are extracted using bounding box information from our cropping tool and have varying widths $(w_c)$ and heights $(h_c)$ proportional to the size of the organism. The resized images $(I_r)$ are adjusted so that the shorter dimension, either width $(w_r)$ or height $(h_r)$, is set to a constant size of 256 px, while the other dimension is scaled proportionally to maintain the aspect ratio, resulting in a dimension greater than 256 px.

We calculated the scale-factor $(f_s)$ as follows:

$$f_s = \frac{\min(w_c, h_c)}{256} \tag{3}$$

If we define the pixel scale as the number of millimeters per pixel, then the pixel scale of the cropped and resized image (`cropped_256`) is equivalent to the pixel scale of the original image multiplied by the scale factor:

$$\text{pixel\_scale}_{\text{cropped\_256}} = \text{pixel\_scale}_{\text{original}} \times f_s \tag{4}$$

Note the pixel scale of the original image remains unchanged during the cropping process, as cropping only involves cutting out areas around the region of interest (the organism) without scaling the image.

The original images were captured using a Keyence VHX-7000 Digital Microscope system imaging system at a resolution of 2880×2160 pixels. These images were then resized to a resolution of 1024×768 pixels to obtain the original images (`original_full`) of the BIOSCAN-5M dataset. Each pixel in the raw images represents a physical space of 2.95 μm by 2.95 μm. Using this pixel scale and the scale factor obtained from Equation 4, we can estimate the size of the object in the real-world.

## S14.2  HDF5 file

To load data efficiently during the training of the CLIBD baseline, we also generated a 190 GB HDF5 file to store images and related metadata from the BIOSCAN-5M dataset. This file is structured to allow rapid access and processing of large-scale data.

At the top level, the file consists of a *group* of the following *datasets* representing different partitions of BIOSCAN-5M. Each partition includes keys or queries for one or all of the splits (pretraining, validation, or test).

For more information or to download the HDF5 file, the instructions are found at the CLIBD GitHub repo: `https://github.com/bioscan-ml/clibd`.

## S14.3  Taxa of unassigned placement

Some taxonomic labels had "holes" in them due to the complexities of the definition of taxonomic labels. Established taxonomic labels for some species can omit taxonomic ranks because there is currently no scientific need to define a grouping at that taxonomic level.

In particular, we found there were 1,448 genera which were missing a subfamily label because their genus had not been grouped into a subfamily by the entomological community. Note that these genera might at some point in the future be assigned a subfamily, if a grouping of genera within the same family becomes apparent.

This situation of mixed rankings creates a complexity for hierarchical modelling, which for simplicity typically assumes a rigid structure of level across the labelling tree for each sample. We standardized this by adding a placeholder subfamily name where there was a hole, equal to "unassigned <Family_name>". For example, for the genus Alpinosciara, the taxonomic label was originally:

```
Arthropoda > Insecta > Diptera > Sciaridae > [none] > Alpinosciara
```

and after filling the missing subfamily label, it became:

```
Arthropoda > Insecta > Diptera > Sciaridae > unassigned Sciaridae > Alpinosciara
```

This addition ensures that the mapping from genus to subfamily is injective, and labels which are missing because they are not taxonomically defined are not confused with labels which are missing because they have not been identified. Furthermore, this ensures that each subsequent rank in the taxonomic labels provides a partitioning of each of the labels in the rank that proceeds it.

## S14.4  Taxonomic label cleaning

The taxonomic labels were originally entered into the BOLD database by expert entomologists using a drop-down menu for existing species, and typed-in manually for novel species. Manual data entry can sometimes go awry. We were able to detect and resolve some typographical errors in the manual annotations, as described below.

**Genus and species name comparison.**  Since species names take the form "<Genus_name> <species_specifier>", the genus is recorded twice in samples which possess species labels. This

redundancy provides an opportunity to provide a level of quality assurance on the genus-level annotations A few samples (82 samples across 13 species) had a species label but no genus label; for these we used the first word of the species label as the genus label. For the rest of the samples with a species label, we compared the first word of the species label with the genus label, and resolved 166 species where these were inconsistent. These corrections also uncovered cases where the genus name was entered incorrectly more broadly, and we were able to correct genera values which were entered incorrectly even in cases where they were consistent with their species labels or had no species labels.

**Conflicted annotations for the same barcode.** We found many DNA barcodes were repeated across the dataset, with multiple images bearing the same barcode. Overall, there were on average around two repetitions per unique barcode in the dataset. It is already well-established that the COI mitochondrial DNA barcode is a (sub)species-level identifier, i.e. same barcode implies same species, and different species implies different barcodes (Moritz & Cicero, 2004; Sokal & Sneath, 1963; Blaxter et al., 2005). Hence we have a strong prior that samples with the same barcode should be samples of the same species. This presents another opportunity to provide quality assurance on the data, by comparing the taxonomic annotations across samples which shared a DNA barcode. Differences can either arise by typographical errors during data entry, or by differences of opinion between annotators.

We investigated cases where completed levels of the taxonomic annotations differed for the same barcode. This indicated some common trends as values often compared as different due to stylistic differences, where one annotation differed only by casing, white-space, the absence of a 0 padding digit to an identifier code number, or otherwise misspellings. We resolved some such disagreements automatically, by using the version more common across the dataset.

The majority of placeholder genus and species names follow one of a couple of formats such as "<Genus_name> Malaise1234", e.g. "*Oxysarcodexia Malaise4749*". Comparing different taxonomic annotations of the same barcode only allows us to find typos where a barcode has been annotated more than once. However, there are of course more barcodes than species and so there may remain some typos which make two samples of the same species with different barcodes compare as different when they should be the same. To address this, we found labels which deviated from the standardized placeholder name formats and modified them to fit the standardized format. Examples of these corrections include adding missing zero-padding on digits, fixing typos of the word "Malaise", and inconsistent casing. In this way, we renamed the species of 6,756 samples and genus of 3,675 across 7,673 records.

We resolved the remaining conflicts between differently annotated samples of the same barcode as follows. We considered each taxonomic rank one at a time. In cases where there was a conflict between the annotations, we accepted the majority value if at least 90% of the annotations were the same. If the most common annotation was less prevalent than this, we curtailed the annotation at the preceding rank. Curtailed annotations which ended at a filler value (i.e. a subfamily name of the format "unassigned <Family_name>") were curtailed at the last completed rank instead. In total, we dropped at least one label from 3,478 records.

Next, we considered barcodes whose multiple annotations differed in their granularity. In such cases, we inferred the annotations for missing taxonomic ranks from the samples that were labelled to a greater degree of detail. In total, we inferred at least one label for 172,895 records. We believe these inferred labels are unlikely to have an error rate notably higher than that of the rest of the data. Even so, we provide details about which ranks were inferred in the metadata field `inferred_ranks` in case the user wishes to exclude the inferred labels. This field takes the following values:

- 0 — Original label only (nothing inferred).
- 1 — Species label was copied. (Sample was originally labelled to genus-level.)
- 2 — Genus and (if present) species labels were copied.
- 3 — Subfamily, and every rank beneath it, were copied.
- 4 — Family, and every rank beneath it, were copied.
- 5 — Order, and every rank beneath it, were copied.
- 6 — Class, and every rank beneath it, were copied.

Table S9: Example species from each species set.

| Species set | Genus | Species | Number of samples | | | |
|---|---|---|---|---|---|---|
| | | | All | Train/ Keys | Val | Test |
| seen | Aacanthocnema | Aacanthocnema dobsoni | 3 | 3 | 0 | 0 |
| | Glyptapanteles | Glyptapanteles meganmiltonae | 65 | 45 | 2 | 18 |
| | Megaselia | Megaselia lucifrons | 699 | 640 | 34 | 25 |
| | Pseudomyrmex | Pseudomyrmex simplex | 378 | 335 | 18 | 25 |
| | Rhopalosiphoninus | Rhopalosiphoninus latysiphon | 148 | 116 | 7 | 25 |
| | Stenoptilodes | Stenoptilodes brevipennis | 16 | 10 | 1 | 5 |
| | Zyras | Zyras perdecoratus | 10 | 6 | 0 | 4 |
| unseen | Anastatus | Anastatus sp. GG28 | 42 | 24 | 6 | 12 |
| | Aristotelia | Aristotelia BioLep531 | 87 | 51 | 13 | 23 |
| | Glyptapanteles | Glyptapanteles Whitfield155 | 11 | 6 | 1 | 4 |
| | Megaselia | Megaselia BOLD:ACN5814 | 24 | 13 | 3 | 8 |
| | Orthocentrus | Orthocentrus Malaise5315 | 39 | 23 | 5 | 11 |
| | Phytomyptera | Phytomyptera Janzen3550 | 14 | 8 | 1 | 5 |
| | Zatypota | Zatypota alborhombartaDHJ03 | 9 | 8 | 1 | 0 |
| heldout | Basileunculus | Basileunculus sp. CR3 | 268 | | | |
| | Cryptophilus | Cryptophilus sp. SAEVG Morph0281 | 55 | | | |
| | Glyptapanteles | Glyptapanteles Malaise2871 | 1 | | | |
| | Odontofroggatia | Odontofroggatia corneri-MIC | 13 | | | |
| | Palmistichus | Palmistichus ixtlilxochitliDHJ01 | 416 | | | |
| | gelBioLep01 | gelBioLep01 BioLep3792 | 16 | | | |
| | microMalaise01 | microMalaise01 Malaise1237 | 13 | | | |

**Non-uniquely identifying species names.** Finally, we noted that some species names were not unique identifiers for a species. Theses cases arise where an annotator has used *open nomenclature* to indicate a suspected new species, e.g. "*Pseudosciara sp.*", "*Olixon cf. testaceum*", and "*Dacnusa nr. faeroeensis*". Since this is not a uniquely identifying placeholder name for the species, it is unclear whether two instances with the same label are the same new species or different new species. For example, there were 1,247 samples labelled as "*Pseudosciara sp.*", and these will represent a range of new species within the Pseudosciara genus, and not repeated observations of the same new species. Consequently, we removed such species annotations which did not provide a unique identifier for the species. In total, 198 such species values were removed from 5,101 samples.

**Conclusion.** As a result of this cleaning process we can make the following claims about the dataset, with a high degree of confidence:

- All records with the same barcode have the same annotations across the taxonomic hierarchy.
- If two samples possess a species annotation and their species annotation is the same, they are the same species. (Similarly for genus level annotations, etc.)
- If two samples possess a species annotation and their species annotations differ, they are not the same species. (Similarly for genus level annotations, etc.)

# S15   Dataset partitioning — Additional details

## S15.1   Species sets

As summarized in §4.1 of the main text, we first partitioned the data based on their species label into four categories as follows:

- *Unknown*: samples without a species label (note: these may truly belong in any of the other three categories).
- *Seen*: all samples whose species label is an established scientific name of a species. Species which did not begin with a lower case letter, contain a period, contain numerals, or contain

"malaise" (case insensitive) were determined to be labelled with a catalogued, scientific name for their species, and were placed in the *seen* set.

- *Unseen*: Of the remaining samples, we considered the placeholder species which we were most confident were labelled reliably. These were species outside the seen species, but the genus occurred in the seen set. Species which satisfied this property and had at least 8 samples were placed in the *unseen* set.

- *Heldout*: The remaining species were placed in *heldout*. The majority of these have a placeholder genus name as well as a placeholder species name, but some have a scientific name for their genus name.

This partitioning ensures that the task that is posed by the dataset is well aligned with the task that is faced in the real-world when categorizing insect samples. Example species for each species set are shown in Table S9, and the number of categories for each taxonomic rank are shown in Table S10.

Table S10: Number of (non-empty) categories for each taxa, per species set.

| Species set | Phylum | Class | Order | Family | Subfamily | Genus | Species |
|---|---|---|---|---|---|---|---|
| unknown | 1 | 10 | 52 | 869 | 1,235 | 4,260 | 0 |
| seen | 1 | 9 | 42 | 606 | 1,147 | 4,930 | 11,846 |
| unseen | 1 | 3 | 11 | 64 | 118 | 244 | 914 |
| heldout | 1 | 4 | 22 | 188 | 381 | 1,566 | 9,862 |
| overall | 1 | 10 | 55 | 934 | 1,542 | 7,605 | 22,622 |

## S15.2 Splits

To construct partitions appropriate for a closed world training and evaluation scenario, we partitioned the seen data into `train`, `val`, and `test` partitions. Because many of the DNA barcodes have more than one sample (i.e. multiple images per barcode), we partitioned the data at the barcode level. The data was highly imbalanced, so to ensure the `test` partition had high sample efficiency, we flattened the distribution for the `test` set. For each species with at least 2 barcodes and at least 8 samples, we selected barcodes to place in the `test` set. We tried to place a number of samples in the `test` set which scaled linearly with the number of samples for the species, starting with a minimum of 4, and capped at a maximum of 25 (reached at 92 samples total). The target increased at a rate of $1/4$. We capped the number of barcodes to place in the `test` set at a number that increased linearly with the number of barcodes for the species, starting at 1 and increasing at a rate of $1/3$. This flattened the distribution across species in the `test` set, as shown in Figures S11e, S12e, and S13e.

Table S11: Number of (non-empty) categories for each taxa, per partition.

| Partition | Phylum | Class | Order | Family | Subfamily | Genus | Species |
|---|---|---|---|---|---|---|---|
| pretrain | 1 | 10 | 52 | 869 | 1,235 | 4,260 | 0 |
| train | 1 | 9 | 42 | 606 | 1,147 | 4,930 | 11,846 |
| val | 1 | 5 | 27 | 350 | 598 | 1,704 | 3,378 |
| test | 1 | 6 | 27 | 352 | 594 | 1,736 | 3,483 |
| key_unseen | 1 | 3 | 11 | 64 | 118 | 244 | 914 |
| val_unseen | 1 | 3 | 11 | 62 | 116 | 240 | 903 |
| test_unseen | 1 | 3 | 11 | 62 | 113 | 234 | 880 |
| other_heldout | 1 | 4 | 22 | 188 | 381 | 1,566 | 9,862 |
| overall | 1 | 10 | 55 | 934 | 1,542 | 7,605 | 22,622 |

To evaluate model performance during model development cycles, we also created a validation partition (`val`) with the same distribution as the `test` set. This was partition was created to contain around 5% of the remaining samples from each of the seen species, by selecting barcodes to place in the `val` partition. To mimic the long tail of the distribution, for each species with fewer than 20 samples and at least 6 samples, and for which one of their barcodes had only a single image, we added one single-image barcode to the `val` partition. This step added 1,766 individual samples from the tail; for comparison, our target of 5% of the samples from the tail would be 1,955 samples.

Table S12: Number of species in common between each pair of partitions.

| | pretrain | train | val | test | key_unseen | val_unseen | test_unseen | other_heldout |
|---|---|---|---|---|---|---|---|---|
| pretrain | 0 | 0 | 0 | 0 | 0 | 0 | 0 | 0 |
| train | 0 | 11,846 | 3,378 | 3,483 | 0 | 0 | 0 | 0 |
| val | 0 | 3,378 | 3,378 | 2,952 | 0 | 0 | 0 | 0 |
| test | 0 | 3,483 | 2,952 | 3,483 | 0 | 0 | 0 | 0 |
| key_unseen | 0 | 0 | 0 | 0 | 914 | 903 | 880 | 0 |
| val_unseen | 0 | 0 | 0 | 0 | 903 | 903 | 878 | 0 |
| test_unseen | 0 | 0 | 0 | 0 | 880 | 878 | 880 | 0 |
| other_heldout | 0 | 0 | 0 | 0 | 0 | 0 | 0 | 9,862 |

Table S13: Fraction of species (%) in common between each pair of partitions, relative to the number of species for the row.

| | pretrain | train | val | test | key_unseen | val_unseen | test_unseen | other_heldout |
|---|---|---|---|---|---|---|---|---|
| pretrain | N/A | N/A | N/A | N/A | N/A | N/A | N/A | N/A |
| train | 0.0 | 100.0 | 28.5 | 29.4 | 0.0 | 0.0 | 0.0 | 0.0 |
| val | 0.0 | 100.0 | 100.0 | 87.4 | 0.0 | 0.0 | 0.0 | 0.0 |
| test | 0.0 | 100.0 | 84.8 | 100.0 | 0.0 | 0.0 | 0.0 | 0.0 |
| key_unseen | 0.0 | 0.0 | 0.0 | 0.0 | 100.0 | 98.8 | 96.3 | 0.0 |
| val_unseen | 0.0 | 0.0 | 0.0 | 0.0 | 100.0 | 100.0 | 97.2 | 0.0 |
| test_unseen | 0.0 | 0.0 | 0.0 | 0.0 | 100.0 | 99.8 | 100.0 | 0.0 |
| other_heldout | 0.0 | 0.0 | 0.0 | 0.0 | 0.0 | 0.0 | 0.0 | 100.0 |

The remaining barcodes with samples of seen species are placed in the `train` partition. For retrieval paradigms, we use the `train` partition as keys and the `val` and `test` partitions as queries.

For the unseen species, we use the same methodology as for the seen species to create and `val_unseen`, with the exception that the proportion of samples placed in the `val_unseen` partition was increased to 20% to ensure it is large enough to be useful. The remaining samples of unseen species are placed in the `keys_unseen` partition. For retrieval paradigms, we use the `keys_unseen` partition as keys and the `val_unseen` and `test_unseen` partitions as queries. For open world evaluation, we train on the `test` partition, without presenting any samples from the unseen species during training, and evaluate on `test_unseen`.

The samples of heldout species are placed in the partition `other_heldout`. The utility of these species varies depending on the model paradigm. In particular, we note that as these species are in neither the seen nor unseen species, they can be used to train a novelty detector without the novelty detector being trained on unseen species.

The samples of unknown species are placed entirely in the pretrain partition, which can be used for self-supervised pretraining, or semi-supervised learning.

To aid comparison between the coverage of the partitions, we show the number of species in common between each pair of partitions (Table S12), and the percentage of species in common (Table S13). This is a block-diagonal matrix as species labels do not overlap between species sets. The `train` partition has higher diversity than the `val` and `test` partitions, which each cover less than 30% of the seen species. This is due to the long-tail of the distribution — of the 11,846 species, 7,919 species (two thirds) have 6 or fewer samples, and of these 3,756 species only have a single sample. However,

Table S14: Number of genera in common between each pair of partitions.

| | pretrain | train | val | test | key_unseen | val_unseen | test_unseen | other_heldout |
|---|---|---|---|---|---|---|---|---|
| pretrain | 4,260 | 2,372 | 1,190 | 1,206 | 217 | 214 | 209 | 682 |
| train | 2,372 | 4,930 | 1,704 | 1,736 | 244 | 240 | 234 | 519 |
| val | 1,190 | 1,704 | 1,704 | 1,517 | 151 | 148 | 145 | 266 |
| test | 1,206 | 1,736 | 1,517 | 1,736 | 157 | 154 | 151 | 276 |
| key_unseen | 217 | 244 | 151 | 157 | 244 | 240 | 234 | 177 |
| val_unseen | 214 | 240 | 148 | 154 | 240 | 240 | 232 | 175 |
| test_unseen | 209 | 234 | 145 | 151 | 234 | 232 | 234 | 172 |
| other_heldout | 682 | 519 | 266 | 276 | 177 | 175 | 172 | 1,566 |

Table S15: Fraction of genera (%) in common between each pair of partitions, relative to the number of genera for the row.

| | pretrain | train | val | test | key_unseen | val_unseen | test_unseen | other_heldout |
|---|---|---|---|---|---|---|---|---|
| pretrain | 100.0 | 55.7 | 27.9 | 28.3 | 5.1 | 5.0 | 4.9 | 16.0 |
| train | 48.1 | 100.0 | 34.6 | 35.2 | 4.9 | 4.9 | 4.7 | 10.5 |
| val | 69.8 | 100.0 | 100.0 | 89.0 | 8.9 | 8.7 | 8.5 | 15.6 |
| test | 69.5 | 100.0 | 87.4 | 100.0 | 9.0 | 8.9 | 8.7 | 15.9 |
| key_unseen | 88.9 | 100.0 | 61.9 | 64.3 | 100.0 | 98.4 | 95.9 | 72.5 |
| val_unseen | 89.2 | 100.0 | 61.7 | 64.2 | 100.0 | 100.0 | 96.7 | 72.9 |
| test_unseen | 89.3 | 100.0 | 62.0 | 64.5 | 100.0 | 99.1 | 100.0 | 73.5 |
| other_heldout | 43.6 | 33.2 | 17.0 | 17.6 | 11.3 | 11.2 | 11.0 | 100.0 |

these rare species only constituted a small fraction of the `train` samples—only 17,572 samples are members of species with 6 or fewer samples, which is 6% of the `train` partition. Due to our selection process for unseen species, in which only species with enough samples to be confident they are accurate are included, a much higher fraction of the unseen species are included in `val_unseen` and `test_unseen`.

Similarly, we show the number and percentage of genera in common between pairs of partitions (Table S14 and Table S15, respectively). We see that the genera across all seen and unseen species set partitions are contained in the `train` partition.

In Figure S14, we show the number of samples per partition. The plot illustrates the vast majority of the samples (91%) are in the pretrain partition, and most samples are only labelled to family level (67%).

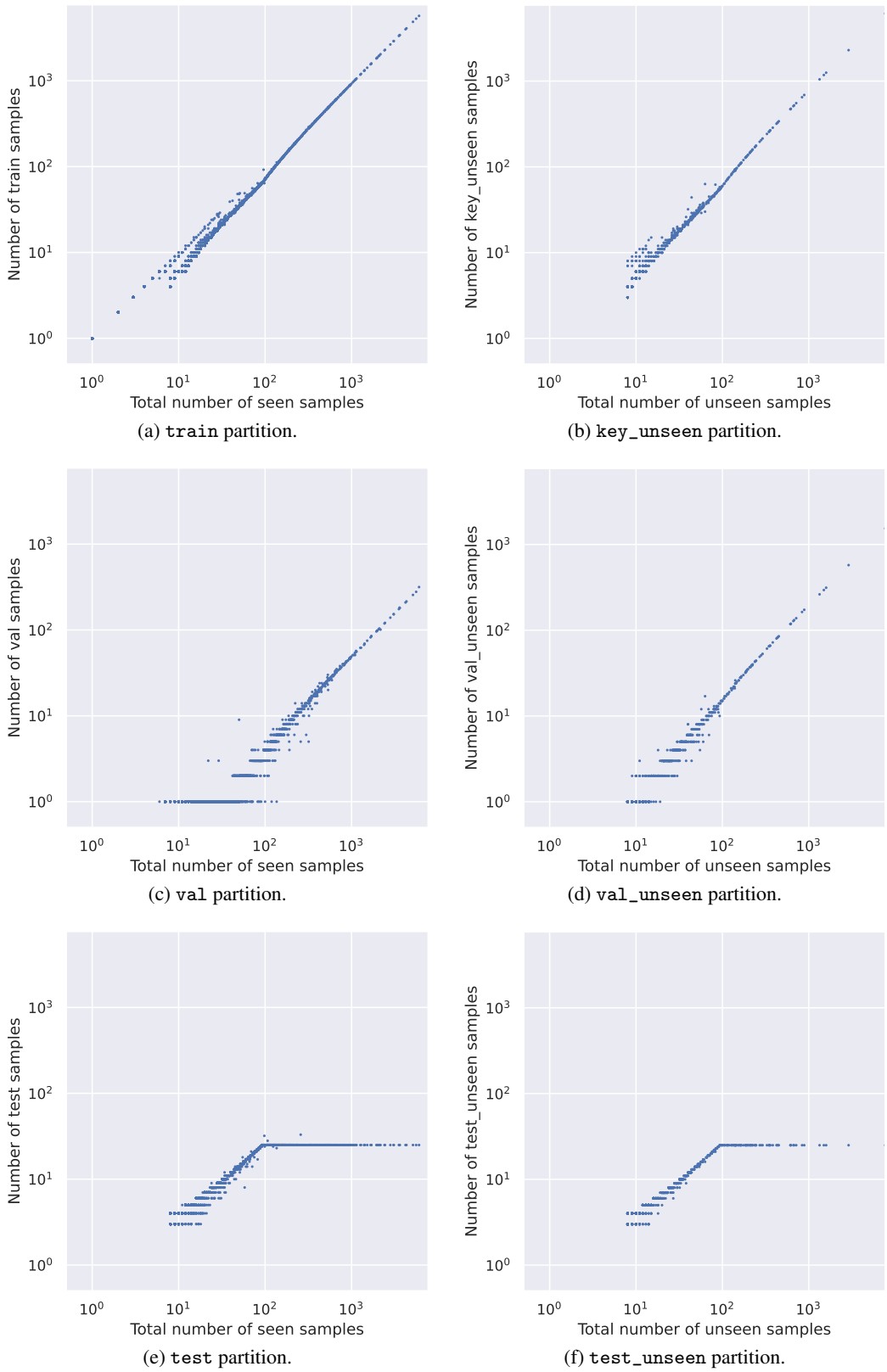

Figure S11: Number of samples in species set and partition, per species.

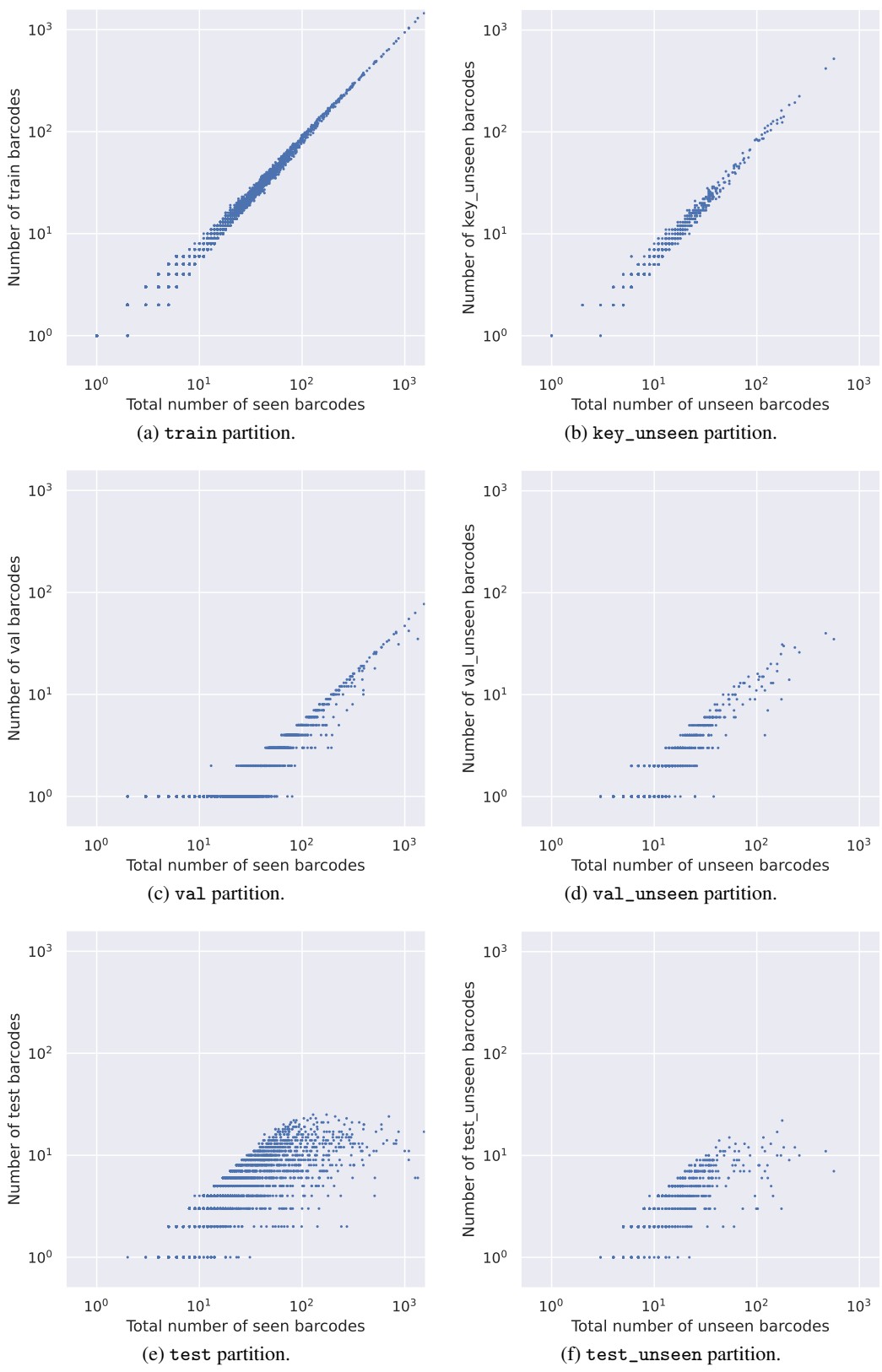

Figure S12: Number of barcodes in species set and partition, per species.

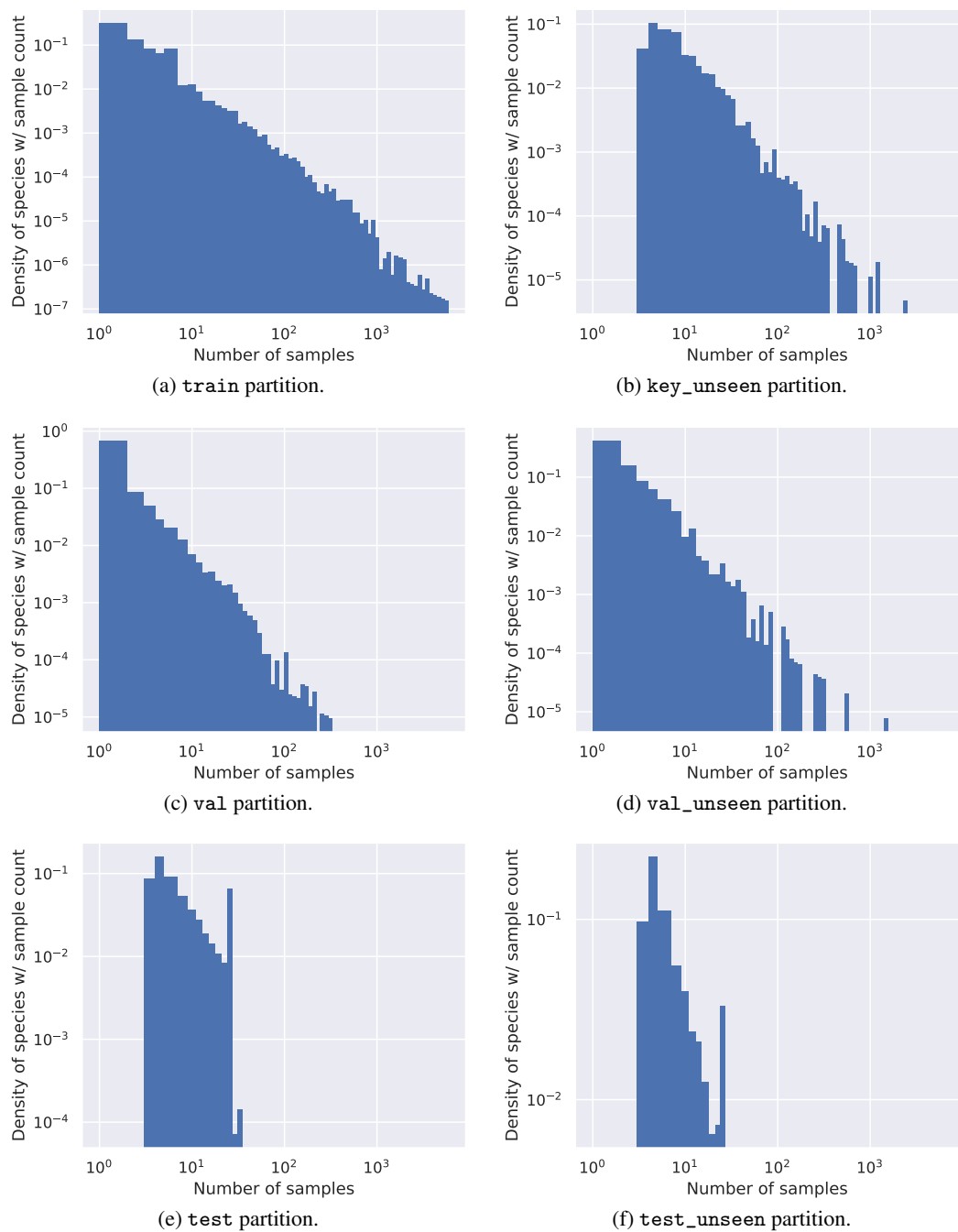

Figure S13: **Distribution of species prevalences across the main data partitions.** Note the log-log axes due to the power law distribution of the data.. The majority of species are infrequent, but some species have many samples. The `train` and `key_unseen` partitions have similar distributions to the overall distribution for *seen* and *unseen* species. The `val` partitions have the same distribution, but shifted left as they they contain a fixed fraction of the samples per species. The `test` partitions are truncated with a minimum and maximum number of samples per species, which flattens the distribution over species for these partitions.

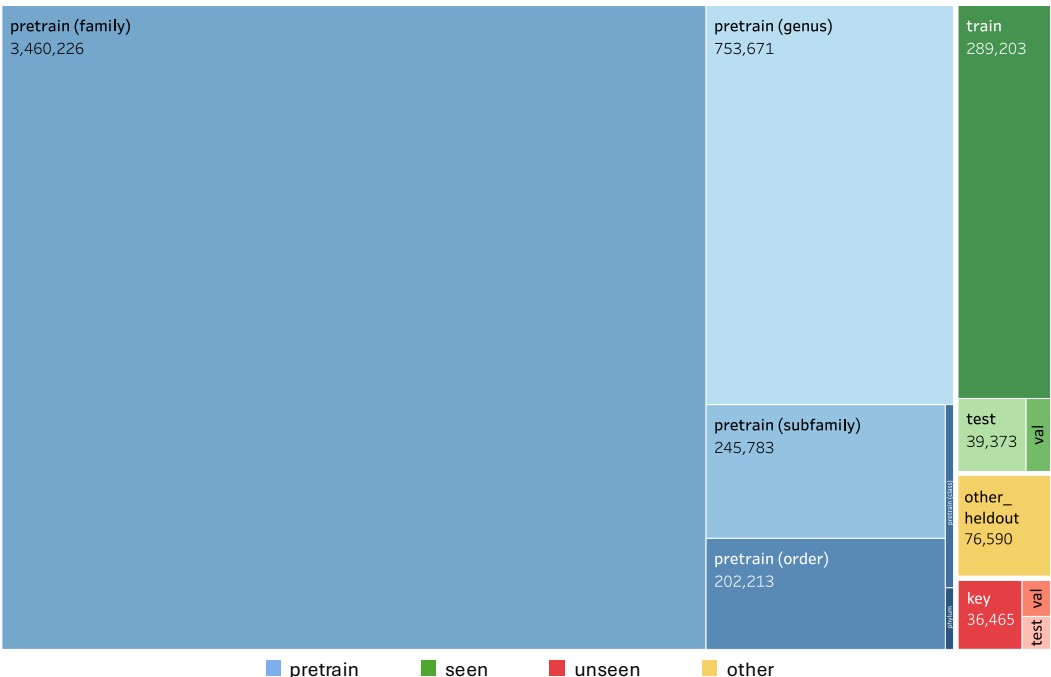

Figure S14: **Treemap diagram showing number of samples per partition.** For the `pretrain` partition (blues), we provide a further breakdown indicating the most fine-grained taxonomic rank that is labelled for the samples. For the remainder of the partitions (all of which are labelled to species level) we show the number of samples in the partition. Samples for seen species are shown in shades of green, and unseen in shades of red.

## S15.3 Distributional shift

As described above, we partitioned our data into sets to use for open- and closed-world tasks. The division of our data was directed by the labels, with scientific names places in the "seen" species set and placeholder names in the "unseen" species set. This partitioning method means our open-world dataset should be, by construction, well-aligned with the open-world task seen in practice for novel data collection. Novel arthropod species are continually being discovered and identified as species that are new to science, and if we assume there is a uniform efficiency for naming across taxa the distribution is of placeholder names is likely to match the distribution of new species discovery. However, this distribution does not necessarily match that of taxa prevalence, due to several factors such as non-uniform speciation rates across arthropods.

We investigated the difference in the distribution at order and class level for the dataset partitions, tabulated in Table S16 and illustrated in Figures S15 and S16. We observe that the Diptera (fly) class of Insecta dominates the overall and `pretrain` dataset (Figure S15b), but "seen" partitions (Figure S15c) have a flatter distribution with more prevalence of two non-Insecta orders—Arachnida (spiders, etc.) and Collembola (springtails)—and more instances of non-Diptera Insecta classes. The distribution is even flatter for the `test` partition (Figure S16e), due to our capped subsampling methodology when creating the partition.

For "unseen" partitions (Figure S15d), we find the data is split nearly equally between three dominant Insecta classes—Diptera (flies), Hymenoptera (bees, ants, etc.), and Lepidoptera (moths, etc.). The `test_unseen` partition (Figure S16f) contains even more Hymenoptera (around 62%). The `other_heldout` partition (Figure S15e) has even less Diptera, and is instead dominated by Lepidoptera and Hymenoptera.

Users of the BIOSCAN-5M dataset should thus be sure to consider the effect of this distributional shift on their results if they wish to make direct comparisons between the `test` performance and the `test_unseen` performance—results for these partitions are not intended to be directly comparable to each other.

Table S16: **Distribution of predominant classes and orders across data splits.** For each taxonomic class present in the dataset, and selected orders which have a prevalence of at least 0.5% for at least one split, we show the proportion of samples in each split (%) bearing this taxonomic label. Values for orders which never occur in a split are left empty. Background: linear colour scale from 0% (white) to 75% (blue).

| Class | Order | pretrain | Seen species | | | Unseen species | | | other_heldout |
| | | | train | val | test | key_unseen | val_unseen | test_unseen | |
|---|---|---|---|---|---|---|---|---|---|
| Arachnida | Araneae | 0.35 | 2.25 | 2.15 | 4.11 | 0.17 | 0.16 | 0.49 | 0.14 |
| | Mesostigmata | 0.08 | 0.13 | 0.14 | 0.36 | 0.49 | 0.53 | 0.81 | 0.16 |
| | Sarcoptiformes | 0.09 | 0.34 | 0.33 | 0.61 | | | | |
| | (Other) | 0.31 | 0.13 | 0.12 | 0.27 | | | | 0.01 |
| Branchiopoda | (Total) | 0.00 | 0.01 | 0.01 | 0.04 | | | | |
| Chilopoda | (Total) | 0.00 | 0.00 | | | | | | 0.01 |
| Collembola | Entomobryomorpha | 0.57 | 2.80 | 2.91 | 1.03 | 0.06 | 0.07 | 0.14 | 0.65 |
| | (Other) | 0.19 | 0.43 | 0.45 | 0.49 | 0.16 | 0.17 | 0.36 | 0.02 |
| Copepoda | (Total) | 0.00 | 0.00 | | | | | | |
| Diplopoda | (Total) | 0.00 | 0.00 | | 0.01 | | | | |
| Diplura | (Total) | 0.00 | | | | | | | |
| Insecta | Coleoptera | 5.02 | 4.47 | 4.20 | 7.44 | 0.39 | 0.43 | 0.94 | 0.48 |
| | Diptera | 73.64 | 60.56 | 61.75 | 49.21 | 38.44 | 38.96 | 21.74 | 10.19 |
| | Hemiptera | 5.06 | 7.75 | 7.51 | 10.15 | 0.18 | 0.12 | 0.36 | 0.05 |
| | Hymenoptera | 10.23 | 11.64 | 11.42 | 16.00 | 37.46 | 36.50 | 62.32 | 41.84 |
| | Lepidoptera | 2.51 | 4.75 | 4.26 | 5.96 | 22.65 | 23.04 | 12.79 | 46.39 |
| | Psocodea | 0.84 | 2.05 | 2.09 | 1.22 | | | | 0.01 |
| | Thysanoptera | 0.20 | 2.02 | 2.04 | 1.77 | | | | 0.00 |
| | Trichoptera | 0.17 | 0.24 | 0.24 | 0.58 | 0.01 | 0.01 | 0.05 | 0.01 |
| | (Other) | 0.38 | 0.31 | 0.30 | 0.66 | | | | 0.06 |
| Malacostraca | (Total) | 0.00 | 0.09 | 0.08 | 0.10 | | | | |
| Ostracoda | (Total) | 0.00 | 0.00 | | | | | | |

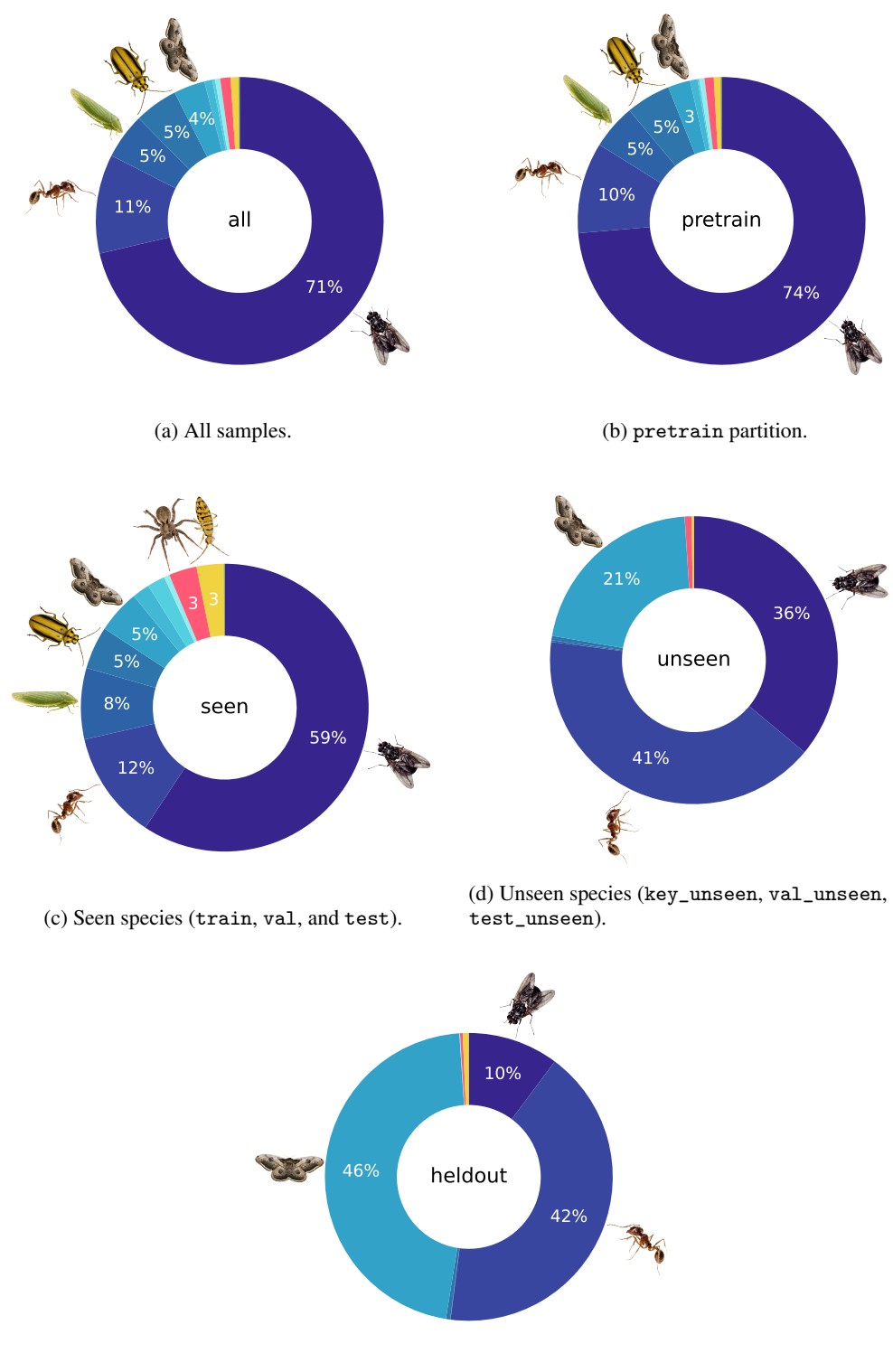

(a) All samples.

(b) `pretrain` partition.

(c) Seen species (`train`, `val`, and `test`).

(d) Unseen species (`key_unseen`, `val_unseen`, `test_unseen`).

(e) `other_heldout` partition.

Figure S15: **Distribution of classes and insect orders.** In each panel, the distribution of taxa is shown for one species set of the dataset. Classes are shown in different hues—Arachnida: red, Collembola: yellow, Insecta orders: shades of blue varying by order, other classes: green. Icons are redistributed under CC BY(-NC) or Canva pro license, respectively. See Table S16 for names and more detailed values.

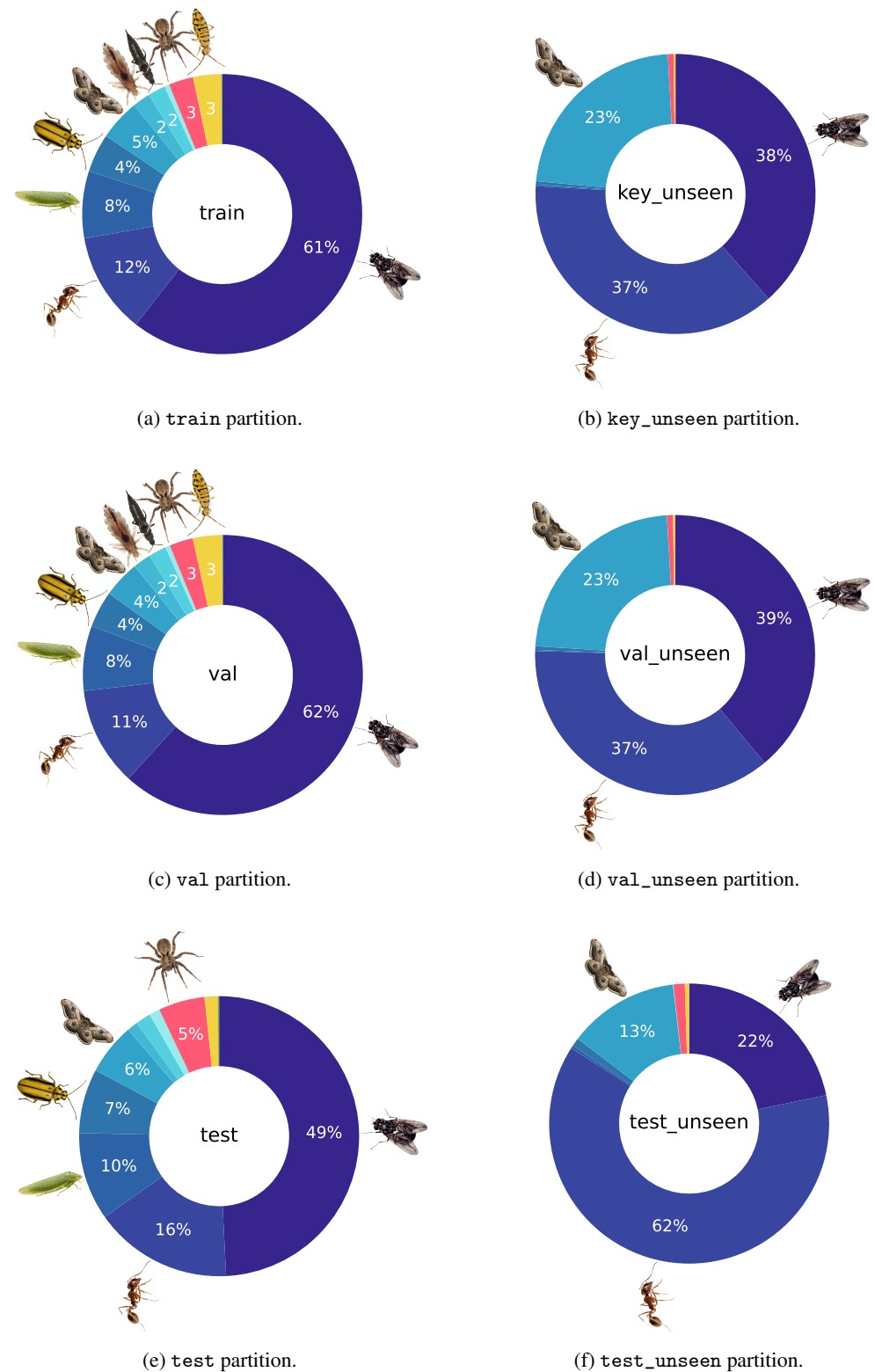

(a) `train` partition.

(b) `key_unseen` partition.

(c) `val` partition.

(d) `val_unseen` partition.

(e) `test` partition.

(f) `test_unseen` partition.

Figure S16: **Distribution of classes and insect orders.** Each panel shows the distribution for one partition. Classes are shown in different hues—Arachnida: red, Collembola: yellow, Insecta orders: shades of blue varying by order, other classes: green. Icons are redistributed under CC BY(-NC) or Canva pro license, respectively. See Table S16 for names and more detailed values.