# OpenReview forum: "BIOSCAN-5M: A Multimodal Dataset for Insect Biodiversity"
_NeurIPS.cc/2024/Datasets_and_Benchmarks_Track — NeurIPS 2024 Track Datasets and Benchmarks Poster_

### Official Review · Reviewer_eL4D · 2024-07-14
**5x larger**

**Rating:** 4
**Confidence:** 3

**Review:**

The paper does a thorough analysis in terms of benchmarking which is very similar to the previous papers like BIOSCAN-CLIP.
The main contribution of the paper is providing 5x size data.

**Strengths:**

size of the data, even though still the data taxonomy coverage/resolution still at the same level.

**Additional Feedback:**

As you can see in your benchmark, adding 5x more data has not significantly improved the model at the genus/species level. Could the lesson be that increasing the volume of data at other ranks does not help in making a better classifier at the genus/species level? It would be helpful to know your future plans/strategy for improving model in this field, given that this is your second release. Would you consider releasing 10x more data in the same manner next year?

**Clarity:**

It is super confusing when you use the term 'species' both as a thing that you have provided data on and as 'a category of living things that ranks below a genus.' I highly recommend using 'taxon/taxa' when referring to something that can be at any rank. For example, one instance is in line 224: 'the task is species-level identification of samples from species that have been seen during the training.' It would have been less confusing to re-write it as 'the task is species-level identification of samples from taxa that have been seen during the training.' Therefore, I suggest renaming 'species set' in the table and considering this change everywhere else in the paper, if you really mean species as a taxonomy level or species as taxa.

**Correctness:**

I wonder why TreeOfLife-10M (introduced by BioCLIP paper) is not listed in the table 1. TreeOfLife-10M consist of "iNaturalist, BIOSCAN-1M, and Encyclopedia of Life".

**Documentation:**

1) seems different model has been used for filtering/preparing data. for example, 1 model for cropping images; would be helpful to know what is the performance of the model or at least identifying what were those 2k+837 that were cropped by human and are free from any error.

2) it is helpful to have an indicator in the data to know whether that instance is annotated by human or machine even if human as verified it manually. So I recommend enriching your data and work with these information as well.

2) description of what DNA barcode are listed. what is the technology behind

**Limitations:**

as the author have acknowledged the taxonomy annotation fall short beyond the family rank; this can severely impact 1) relevancy of the sequence classification using advanced ML model. classification at such level can effectively be done, using conventional bioinformatics tools like sequence similarities. 2) real world impact on biodiversity - whether it is helpful to be able to identify taxa at family level

also, it is not clear if the DNA barcode is unique enough to classify these taxa at species level or not. As I said, is helpful to understand what technology is behind DNA barcode and how it is curated.

**Opportunities For Improvement:**

size & geographical information are not used in any of the benchmark. Since bioscan-5m contribution is providing these additional data, it would have been helpful to see how and if leveraging these information will benefit the model at species / genus level.

Perhaps, adding an error analysis to your benchmark can make it more differentiable from benchmarks on BIOSCAN-CLIP; e.g discussing if there is any pattern for error or confusion for the model. that can guide others, on effectively collect data that can complement yours; e.g. how does the error correlate with size / geography inforamtion; do we need more picutres from small insects ? how well we in central America (more coverage on taxa) comparing with other region of the world. My point is providing a suggestion/direction to the field (data on insect) for making better future benchmark/data.

**Relation To Prior Work:**

it is clearly discussed and in fact very similar to how other papers in the field (e.g. BIOSCAN-CLIP) has benchmarked.

**Summary And Contributions:**

the paper is continuation of "A Step Towards Worldwide Biodiversity Assessment: The BIOSCAN-1M Insect Dataset" and the "BIOSCAN-CLIP" from last year. Major distinguish is including taxonomical annotation, along with meta data.

---

> ### Author Rebuttal · Authors · 2024-08-16
>
> We thank the reviewer for their feedback and helpful suggestions toward improving the paper's clarity.
>
> **RE: Error analysis**
>
> We agree further error analysis can provide insights into challenges and potential directions for data collection. Our experiments show the usefulness of including DNA barcodes, and the need for more data on underrepresented species to improve fine-grained taxa classification. We will add additional analysis to §4.4 investigating where the model fails, and details of which families and geographic locations perform poorly.
>
> **RE: Annotations fall short beyond family**
>
> Indeed, many samples are labelled only to family-level as it is very challenging and labour-intensive to provide genus/species labels. By sharing BIOSCAN-5M, we hope tools will be developed which enable finer-grained classification from images and DNA barcodes.
>
> - **Relevancy of advanced ML models for sequence classification.** While conventional bioinformatics tools can do a reasonable job at classifying DNA sequences to family rank, there is room for improvement. Moreover, we are also interested in classifying images lacking barcodes.
> - **Usefulness of family-level classification.** In BIOSCAN-1M, the main experimental scope was image classification at order and family ranks. The AI tool proposed in B-1M is now used by the Centre for Biodiversity Genomics (CBG) to provide automated taxonomic classification at these ranks. These broad categories are already very useful to provide information on organisms that would otherwise not be classified at all.
> - **Sparse labels for fine-grained classification.** Our experiments show we can leverage unlabelled data for BERT and CLIP-style pretraining. These models then enable sequence/multi-modal matching against a sparse set of samples labelled at genus/species level.
>
> **RE: Are DNA barcodes unique enough to classify species**
>
> DNA barcodes were developed for the goal of species identification. They serve as species identification tools since sequence variation across species is higher than within a species. It is true that this becomes less applicable for higher taxonomic ranks.
>
> **RE: Tech behind DNA barcodes**
>
> Sequences used here are part of a global reference library (BOLD [3]) and were generated following established standard protocols [1,2]. We will add these additional references to the paper.
>
> **RE: TreeOfLife-10M**
>
> We omitted TreeOfLife-10M because it is a metadataset, but appreciate the feedback and will add it.
>
> **RE: Overloading "species"**
>
> Thank you, we will adjust so as to not overload words.
>
> **RE: Cropping model**
>
> The image cropping model used in BS-5M is an improved version of the one we introduced in BS-1M, retrained on a combination of its original training data and newly annotated BS-1M images we identified as incorrectly cropped. We compare the two models on an eval set of 100 additional images that also weren't well cropped and find the retrained model performs better on these. See Supp. Mat. §S11.1 for details. If needed, we would be happy to also annotate some BIOSCAN-5M images that were not in BS-1M and add the performance on those to the paper.
>
> **RE: Indicate if annotation is human, machine, or verified by human.**
>
> Tracking this is indeed very helpful but was only recently implemented at CBG, so unfortunately it is not available for this dataset.
>
> **RE: 5x more data hasn't significantly improved the model**
>
> - In Tables 5 & 8, we compare contrastive training to align image and DNA with the BS-1M pretrain split established in BIOSCAN-CLIP vs our BS-5M pretrain split. We also notice using more data did not improve the model at within-modality retrieval (DNA→DNA, image→image). However, **more data did improve the cross-modality retrieval** (image→DNA) for seen species. We suspect that using more data helped pull the images and DNA representations together, but didn't improve the DNA or image representations individually. This is in line with the cross-modal contrastive loss function we used. In addition, LoRA fine-tuning may limit what the model can learn. We plan to explore models that also include same-modality SSL, and full fine-tuning of the models. Also, note the comparison with BS-1M is not purely a difference of scale, but also due to the difference in how the data split is done—the species distribution is not the same.
> - In addition to the larger scale of the data, BS-5M covers a wider range of genera and species. The increased diversity allows us to measure the performance in a more challenging setting.
>
> **RE: Future plans**
>
> BIOSCAN's sampling is already very much driven by trying to increase diversity in the dataset. Locations are picked on this premise, so we will add more sites at different locations, aiming to collect specimens from as many regions of the world as possible.
>
> We are using only one collection method; insect orders will not change as we have collected all known ones. We will increase the amount of records in each group at a steady rate for each. Some taxonomic groups naturally have lower species diversity, and will unfortunately remain challenging to study comprehensively for a long time (even with 10M images).
>
> To improve model performance we'd ideally want more samples of rare species, but selecting for rare species is by definition not really possible, esp. with standardized methods. There are no plans to refocus on labelling existing unlabelled specimens simply because it is so time consuming and not scalable, but we do anticipate models built from BIOSCAN-5M may aid future AI-assisted annotation.
>
> **Refs**
>
> [1] Hebert et al. 2018. A sequel to Sanger: amplicon sequencing that scales. BMC Genomics 19:219. doi:10.1186/s12864-018-4611-3
>
> [2] deWaard et al. 2008. Assembling DNA Barcodes: Analytical Protocols. doi:10.1007/978-1-59745-548-0_15
>
> [3] Ratnasingham, Hebert. 2007. BOLD: The Barcode of Life Data System. Mol Ecol Notes. 1;7(3):355-364. doi:10.1111/j.1471-8286.2007.01678.x

---

> ### Comment · Reviewer_eL4D · 2024-08-29
> **still scalling the data is the primary contribution of the work**
>
> I am uncertain about the relevance of providing sequence data annotated at the family level when the primary goal is performance evaluation at this level, which involves approximately 600 unique categories (number of insect families).
>
> Let’s revisit the stated goal from the abstract of the paper:
>
> "First, ...we  demonstrate the impact of using this large reference library on species- and genus-level classification performance."
> Let’s revisit the stated goal from the abstract of the paper:
>
> In response to the "Usefulness of family-level classification," it is mentioned that the goal is actually to predict at the family level. However, I feel that the abstract does not accurately reflect this focus on the family level as the primary outcome and goal. To clarify, I am not criticizing the lower performance at this level. The "Data and Benchmark" session for NeurIPS is primarily intended to present unique data and benchmarks. My concern is that if the goal is indeed to make predictions at the family level, it is unrealistic to expect results at lower taxonomic levels when labeled data is unavailable - so it is hard-limitation of the provided data.
>
> I am still doubtful whether this data attract future models yielding important insight.
>
> In short, the only contribution I see is the scaling up of the data from 1M to 5M. While this is undoubtedly a significant amount of work, I don't believe it aligns with the scope of the NeurIPS Data and Benchmark track. Considering that a similar dataset was published by this group last year, I believe it would be fair to pass the opportunity this year to a group presenting a more unique dataset.

---

> > ### Author Response · Authors · 2024-08-29
> > **Response to Reviewer eL4D addressing "scalling the data is the primary contribution of the work"**
> >
> > > In response to the "Usefulness of family-level classification," it is mentioned that the goal is actually to predict at the family level.
> >
> > We would like to clarify that the **primary goal of our work (BIOSCAN-5M) is *not* performance evaluation at the family level**. In our rebuttal, we discussed the usefulness of family-level classification solely in response to the reviewer’s question regarding its real-world impact on biodiversity and whether classifying taxa at the family level is beneficial. We explored family level classification in our previous paper, BIOSCAN-1M [1], but in BIOSCAN-5M we focus on genus and species level classification, as per our abstract. We apologize for any confusion our rebuttal may have caused on this point.
> >
> > > it is unrealistic to expect results at lower taxonomic levels when labeled data is unavailable - so it is hard-limitation of the provided data.
> >
> > Note that the number of samples labelled to genus level in BIOSCAN-5M is *larger* than the total number of samples in BIOSCAN-1M. Hence we are optimistic that the quality of model we attained at classifying data at family level from BIOSCAN-1M can be attained at genus level with BIOSCAN-5M, even though only a quarter of the samples are labelled to genus-level.
> >
> > **BIOSCAN-5M Contributions**
> >
> > In case they have not seen it, we would like to kindly direct the reviewer to our [rebuttal overall response](https://openreview.net/forum?id=A33u66KmYf&noteId=GeANYH619o
> > ) to all reviewers, where we outlined the distinctions between BIOSCAN-1M and BIOSCAN-5M. To further clarify, we would like to make the following points:
> >
> > - Our **primary focus is fine-grained taxonomic classification down to the genus and species levels**.
> >
> > - We have released a data split for both **closed-world** and **open-world settings**, utilizing data labelled at the species level for evaluation while reserving unlabelled data for pretraining. We believe that our proposed splitting mechanisms and dataset splits offer significant benefits to the ML community.
> >
> > - **All experiments** we conducted for BIOSCAN-5M include evaluation down to the **species level**.
> >
> > - In addition to expanding the dataset, a key objective is to benchmark three distinct tasks, where each task comes with varying levels of difficulty. This is **to demonstrate BIOSCAN-5M’s multimodal capabilities in real-world experiments**: These tasks are:
> >
> >    - 1. Fine-grained taxonomic classification using DNA sequences.
> >    - 2. Fine-grained taxonomic classification using DNA sequences, images, and textual taxonomic labels.
> >    - 3. Clustering of learned DNA and image embeddings.
> >
> > 	In particular, note that taxonomic classification from images is much more challenging than from DNA barcodes (as demonstrated by our clustering experiments), hence paired data can be very valuable even when it is unlabelled.
> >
> > - Data that is not labelled down to the species level is still valuable for **pretraining**. We emphasize that **unlabeled data** plays a crucial role in pretraining models. In BIOSCAN-5M, we use BERT-style masked sequence modelling to pretrain and encode DNA sequences, coupled with contrastive learning for aligning image and DNA embeddings. By including such pretraining, we enhance the model's ability to generalize across various applications.
> >
> > - Furthermore, we provide **additional information such as size and geographic location**. Although these attributes are not utilized in our experiments, their inclusion enables the research community to explore their potential in developing better models or defining new tasks, such as size or location prediction.
> >
> > **Reference**
> >
> > [1] Gharaee, Z., et al A step towards worldwide biodiversity assessment: The BIOSCAN-1M insect dataset. In Advances in Neural Information Processing Systems, volume 36, pp. 43593–43619.

---

> > > ### Comment · Reviewer_eL4D · 2024-08-30
> > > **Thanks for the clarification**
> > >
> > > Thanks for the clarification on the goal and distinction with the earlier version. It is clear now that the goal is primarily at lower level where ~23% of the data are labeled. Also the distinction with the version from the last year.

---

### Official Review · Reviewer_CH9G · 2024-07-23
**Insects: They are Legion**

**Rating:** 8
**Confidence:** 4
**Correctness:** Benchmarks provided are appropriate.
**Clarity:** The paper is clear and well written.

**Review:**

This is a good paper and I recommend it for the conference. It is an interesting multi-modal task, and it demonstrates some good baseline results which take advantage of the multimodality. The main concern I have is that the work may be too similar to the previous BIOSCAN-1M paper from the same group.

**Strengths:**

The most interesting aspect of the paper is clearly addressing the problem of handling unknown species in insect classification by providing both 'closed-world' and 'open-world' model evaluation tracks. The closed-world tasks resemble standard classification tasks with heldout data, while the open-world task is zero-shot clustering, evaluated using a mutual-information metric for the model clustering compared to the groundtruth label partition.

The paper also provides an interesting case of multimodality, including both DNA snippets and images of insects. This provides a fertile ground for CLIP-style training, and a cross-modal Image-to-DNA retrieval experiment is performed.

**Additional Feedback:**

None

**Documentation:**

The dataset is very well documented and accessible. There is an extensive github page, and the data is provided on a numbe rof platforms (google drive, zenodo, and more).

**Ethics:**

Nope, seems good to me.

**Limitations:**

Yes, they have adequately documented limitations of the work.

**Opportunities For Improvement:**

If the authors could explain more precisely the differences from the previous BIOSCAN-1M work, it will help the reviewers judge the novelty of this work.

**Relation To Prior Work:**

Expands on previous BIOSCAN-1M paper. Clearly situates itself in the wider field, and provides comparison to existing DNA classification models.

**Summary And Contributions:**

The paper contributes a new dataset BIOSCAN-5M for insect biodiversity. The dataset combines DNA barcode observations and image data, and expands on a previously existing BIOSCAN-1M. The paper provides good baseline model scores. Classification is across the full taxonomic hierarchy, with placeholders for species identification for novel/unknown species.

---

> ### Author Rebuttal · Authors · 2024-08-16
>
> We graciously thank the reviewer for the favourable review, for highlighting that the work addresses the problem of unknown species classification through the closed-world and open-world tasks, and noting the scope afforded by the multimodality of the data.
>
> **RE: Stating precisely the differences between BIOSCAN-1M and BIOSCAN-5M**
>
> We appreciate the reviewer’s valuable suggestion to provide a more detailed comparison between BIOSCAN-1M and BIOSCAN-5M. We direct the reviewer to the overall response, where we have addressed this concern.

---

> > ### Comment · Reviewer_CH9G · 2024-08-21
> >
> > Thanks; I have seen the response, and will keep my score as-is.

---

### Official Review · Reviewer_PFXf · 2024-07-24
**A substantial contribution over previous work**

**Rating:** 7
**Confidence:** 3

**Review:**

This paper appears to be well written and the benchmarks are well designed, with minor areas for improvement. The dataset appear to be constructed in a sound manner with the short coming that not all geographic regions are well-covered.

I also like the inclusion of the geographic location and the size information of the organism which was not included in other prior work.
The benchmark now include BERT based models for DNA-DNA/Taxonomy classification, and the inclusion of the zero-shot results improves the usability of the dataset. Overall, I see the dataset as a substantial contribution over BIOSCAN-1M and would be a helpful resource for researchers working with insect, and for biologists looking to employ machine learning methods.

It would be helpful to see if including the habitat and size information would improve the the top-1 accuracies of the classification. The authors may want to consider including this in the benchmarking of future versions of this dataset.

Overall, I feel that this submission is a substantial contribution over the authors' previous work with BIOSCAN-1M and would be a useful resource for ML researchers working with DNA sequence, image classification and biologists looking to employ machine learning methods to their work.

**Strengths:**

The dataset appears to be solid, and well constructed, and the benchmark is quite extensive. I also like the inclusion of the geographic location and the size information of the organism which was not included in other prior work.
The benchmark now include BERT based models for DNA-DNA/Taxonomy classification, and the inclusion of the zero-shot results improves the usability of the dataset. Overall, I see the dataset as a substantial contribution over BIOSCAN-1M and would be a helpful resource for researchers working with insect, and for biologists looking to employ machine learning methods.

**Additional Feedback:**

Thank you for the contribution. I do hope that the authors take in consideration suggestions for improvements in future versions of the dataset.

i/e. Extending the geographical coverage to include northern Africa and Asia, and to
include the habitat and size modality in the classification benchmarks.

**Clarity:**

The paper is very well written and results are presented clearly. I can follow the experimental procedure and the evaluation methods with very little trouble.

**Correctness:**

The benchmarks are well designed, with minor areas for improvement, and I am inclined to accept the author's claims and results as presented. The dataset appear to be constructed in a sound manner.

**Documentation:**

The dataset is well documented, with URL links to the dataset and github included with the paper. There is sufficient detail available to reproduce the author's results

**Ethics:**

No, I do not foresee or suspect there are any ethical concerns with the submission of the dataset.

**Limitations:**

As mentioned above, the geographical coverage of the dataset is limited especially in northern Africa and Asia. This would be a good thing to consider for future versions of the dataset, especially when many insects in the equatorial regions are disease vectors and researchers involved with pathogens would benefit greatly from the work of the authors.

**Opportunities For Improvement:**

I notice the geographical coverage of the dataset is limited especially in the northern Africa and Asian region. However I recognize that this is logistics and potentially biosecurity issue that can be quite complicated to solve.

I might have missed it, but am I correct that the size and habitat is not included with the text encoder in table 5? It would be interesting to see if including the habitat and size information would improve the the top-1 accuracies of the classification. The authors may want to consider including this in the benchmarking of future versions of this dataset.

**Relation To Prior Work:**

The authors made comparison to prior work and demonstrated that the work is a substantial contribution over previous work. If there is one short coming is the benchmark not showing how including modality such as size and geographic location improve the classification performance.

**Summary And Contributions:**

This paper presents BIOSCAN-5M, an expansion upon the BIOSCAN-1M dataset by including over 5 million and over 320,000 categories of arthropod specimens with multi-modal data inputs such as high-resolution images, DNA barcode sequences, taxonomic labels, and geographical information and size information. The authors showcased the applicability of the dataset for machine learning through several benchmark experiments including DNA and image based classification across various taxonomic level classification.

---

> ### Author Rebuttal · Authors · 2024-08-16
>
> We thank the reviewer for their feedback, for describing the paper as well written and benchmarks well thought out, and noting the substantial contribution over BIOSCAN-1M.
>
> **RE: Geographical coverage is limited in northern Africa and Asia**
>
> As noted in our limitations section (Supplementary Materials, S10.3), BIOSCAN-5M has a relatively limited geographic coverage. This constraint arises from several factors, including logistical challenges in accessing secure data collection sites (as the reviewer acknowledges in their review), and funding limitations.
>
> We would also like to note that the BIOSCAN-1M dataset featured data from only three different countries (Costa Rica, South Africa and Canada). In BIOSCAN-5M, we have extended this to 47 different countries spanning to 1650 unique collection sites, so it is a great improvement in the geographic range of data nonetheless. We will continue to endeavour to extend the geographical extent of the data in the future.
>
> **RE: Size and habitat is not included with the text encoder**
>
> This is correct. Please see our overall response, where we discuss this. We will consider including this in the benchmarking in future versions of the dataset, as the reviewer recommends.

---

> > ### Comment · Reviewer_PFXf · 2024-08-17
> >
> > Thank you, I acknowledge the rebutal, and would like to keep my rating as is. Hoping the authors would be able address some of the concerns brought up in the next version of BioSCAN

---

### Official Review · Reviewer_aHng · 2024-07-24

**Rating:** 8
**Confidence:** 5
**Clarity:** The paper is well written.

**Review:**

Pros
- The dataset contains valuable multi-modal annotations.
- The dataset is constructed with meticulous effort, ensuring high-quality annotations and data integrity.
- BIOSCAN-5M is larger-scale and more diverse than BIOSCAN-1M.
- The authors provide a concise introduction of adequate background knowledge related to the biodiversity field.

Cons
- More insights from three benchmarks are appreciated.
- Geographical information has not been leveraged in the benchmarks.

**Strengths:**

The authors provide multi-modal annotations in the dataset, especially DNA barcodes, which are difficult to acquire in general. The work also includes geographical information, which is useful for identifying species.

The Linnaean taxonomy annotations are high-quality, verified with DNA barcode information. Expert annotators review any discrepancies between Linnaean taxonomy prediction and barcode mapping. The paper post-processes the annotations in a cautious way, resolving issues of non-uniquely identifying species names (the same name is assigned to multiple species) and conflicted annotations for the same barcode (the same barcode is mapped to multiple taxonomy). Such a meticulous annotation process establishes a standardized mapping between barcode and Linnaean taxonomy.

Samples in BIOSCAN-5M were collected across 47 countries, much more than those in BIOSCAN-1M across 3 countries, which are more representative of global biodiversity.

The authors provide a concise introduction of adequate background knowledge related to the biodiversity field, covering its significance, real-world applications and methodology. The paper greatly helps bridge the gap between the ML community and the biodiversity field.

**Additional Feedback:**

Could you introduce how barcodes differ from each other across the species, as well as the genera, allowing the readers to understand the difficulty of the open-world setting in DNA-based taxonomic classification?

Could you give a short paragraph description of Linnaean taxonomy as BIOSCAN-1M does to be more self-contained?

From L127-128, the discrepancies between Linnaean taxonomy (I assume the most specific rank is order level) from the model and DNA barcode are reviewed by human experts. What is the ratio of the discrepancies?

Please add a column of genus in Table 3 for the open-world setting.

For multimodal retrieval learning, please explicitly list the category numbers of barcodes for pretraining on BIOSCAN-1M and BIOSCAN-5M.

**Correctness:**

The batch size of 800 for contrastive learning is relatively small. Have you tried larger batch sizes like 10K? Please provide more justification.

**Documentation:**

The work is well documented.

**Limitations:**

The authors discuss the limitations in Supplementary Materials, mentioning the difficulty of accessing ground-truth labels in lower ranks and the sampling bias inherent in specimen collection.

I understand that the assignments to family and lower ranks rely entirely on human expertise due to insufficient data for training a classifier. However, is it possible to train an unsupervised species-level clustering model on barcodes with high accuracy? It allows human experts to focus on examining those boundary barcodes paired with several image samples.

**Opportunities For Improvement:**

More insights from three benchmarks will be appreciated. For example, from Figure 4, could we assume a lower rank has more similar arthropod barcodes? Could we infer that Arthropod taxonomy at family rank is the most associated with visual features?

I would like to see the usage of location information, but it is also acceptable that the paper has not leveraged location information yet.

**Relation To Prior Work:**

Prior work is adequately discussed.

It is grateful that the authors provide a summary table of fine-grained and long-tailed biological datasets.

**Summary And Contributions:**

The authors have curated the BIOSCAN-5M dataset, comprising 5 million high-resolution microscope images of arthropod specimens. The authors and the hired experts have annotated the large-scale images with multi-grained taxonomic ranks, DNA barcodes and geographical information, making the dataset a significant resource for both the biological and ML communities. The authors propose three benchmark experiments to demonstrate the utility of the dataset: DNA-based taxonomic classification; zero-shot image/barcode feature clustering; multimodal retrieval among images, barcodes and Linnaean taxonomy text sequences.

---

> ### Author Rebuttal · Authors · 2024-08-16
>
> We thank the reviewer for their feedback, and for noting the care we took in the construction of the dataset with regards to data acquisition and post-processing.
>
>
> **RE: Could we assume a lower rank has more similar arthropod barcodes?**
>
> Yes, that is correct. We can safely assume that a lower rank has more similar barcodes.
>
> **RE: Could we infer that Arthropod taxonomy at family rank is the most associated with visual features?**
>
> While the question about whether Arthropod taxonomy at the family rank is the most associated with visual features is intriguing, we do not have definitive evidence to confirm this. The association between taxonomy and visual features can be complex and may vary depending on the specific characteristics of the dataset and the methods used. Further analysis and research would be needed to draw more conclusive insights.
>
> **RE: Possibility of training an unsupervised species-level clustering model on barcodes with high accuracy**
>
> It is correct that the assignments to family and lower ranks rely entirely on human expertise, although BINs are species-level clusters based on DNA sequences. Note that BINs are species proxies obtained from clustering DNA barcodes based on sequence similarities without applying conventional ML training methods. We have utilized human expertise to focus on examining boundary barcodes paired with multiple image samples.
>
> **RE: Batch size of contrastive learning**
>
> We selected a batch size of 800 for fair comparison against the prior BIOSCAN-CLIP work on the BIOSCAN-1M dataset. We would have liked to experiment with significantly larger batch sizes, as the reviewer suggests, but this was not possible due to limitations in the capacity of the computational resources we could access for the paper. We did identify an overall trend in experiments in the BIOSCAN-CLIP paper that larger batch sizes can improve taxonomic classification accuracy. We hope that others with the requisite resources may be able to further scale this experiment to its limits, to maximize the benefits of this training and model architecture.
>
> **RE: Introduce how barcodes differ from each other across species/genera to aid reader understanding**
>
> Yes, we will add a description of this to help orient readers unfamiliar with DNA barcodes. Thank you for the suggestion. We will also add the statistics of the amount of variance between barcodes at various taxonomic ranks of the BIOSCAN-5M to the manuscript, which will help support the description quantitatively.
>
> **RE: Add a short description of Linnaean taxonomy**
>
> We will add the description of Linnaean taxonomy to the paper. Thank you for the suggestion.
>
> **RE: What is the ratio of the discrepancies between Linnaean taxonomy**
>
> The frequency of discrepancies was 2%.
>
> **RE: Adding a column of genus in Table 3 for the open-world setting**
>
> This information is currently in the Supplementary Materials (Table S10), but as there is sufficient space to make Table 3 in the main paper wider, we agree it makes sense to add the genus counts here too. Thank you for the suggestion.
>
> **RE: List the category numbers of barcodes for pretraining on BIOSCAN-1M and BIOSCAN-5M (multimodal retrieval learning):**
>
> The number of unique barcodes used for the pretraining stage on each dataset was as follows:
>
> BIOSCAN-1M:
> - Unique barcodes: 526,663
> - Unique BINs: 83,822
>
> BIOSCAN-5M:
> - Unique barcodes: 2,402,283
> - Unique BINs: 315,477
>
> As suggested by the reviewer, we will add this to the text in Section 4.4.

---

> > ### Comment · Reviewer_aHng · 2024-08-31
> > **Thanks for the rebuttal**
> >
> > I've read the rebuttal and the other reviews. I remain quite positive about this work and I believe it's a good contribution to an increasingly important area at the intersection of AI and biology. I will keep my score as is.

---

### Author Rebuttal · Authors · 2024-08-16

We would like to sincerely thank all reviewers for their positive feedback on the paper.

A couple of points were raised by multiple reviewers, which we respond to here.  We will incorporate reviewer feedback, and add clarifying details and discussion to the paper.

**RE: Additional comparison between BIOSCAN-1M and BIOSCAN-5M** (CH9G, eL4D)

The key differences between BIOSCAN-1M and BIOSCAN-5M are as follows:

- **More data**: 5x as many samples
- **More diverse data**: collected from a wider range of geographic locations (BS-1M: 3 countries; BS-5M: 47 countries), and spanning a wider range of insect life (BS-1M: 1 class, 16 orders; BS-5M: 10 classes, 55 orders)
- **A new post-processing step**: The taxonomic labels went through a data cleaning pipeline to discover and address inconsistencies which were present in the original data. Consequently, BS-5M labels are more reliable than those in BS-1M.
- **Geographic location and specimen size information are available** in BIOSCAN-5M (not present in BS-1M).
- **The BIOSCAN-5M partitioning provides comprehensive support for both closed-world and open-world tasks**, whereas BS-1M only provided closed-world partitioning.
- **Our benchmarking experiments also differ from BS-1M**. In BS-1M we provided a baseline with an *image-only* model evaluated at order and family ranks. In BS-5M, we provide three baselines to utilize the *multi-modal* aspects of the dataset (including DNA barcode sequences, textual taxonomic labels and RGB images) and correspondingly explore performance in both closed- and open-world settings.

We will update the text to better capture these differences.

**RE: Lack of utilization of geographic (and size) information in models** (aHng, PFXf, eL4D)

In BIOSCAN-5M, we utilize the biological (taxonomic labels), and genetic (DNA barcode sequences and BIN) information for fine-grained taxonomic classification, but chose to exclude the geographic and size data in our experiments.

Our rationale for this was as follows. Geographic and size information is useful in order to rule out many species that the sample cannot be (e.g. knowing the sample was collected in North America means it can not be a member of a species known to not inhabit North America; knowing the sample is large means it can't be a species that does not grow that large; etc.). Yet even when taken together, the sample's size and collection site clearly do not provide sufficient information to meaningfully classify its species. Meanwhile, the image and/or genetic data *can* be sufficient information to provide an accurate prediction at the species level. As such, we note that any useful models to use geographic/size data will do so in conjunction with image/genetic data, and thus models which only use image/genetic data still serve as a baseline for models which use image/genetic data + geographic/size data.

Since adding geographic/size data to the models is non-trivial, we chose to prioritize establishing a breadth of image/genetic baselines in this work, and leave adding geographic/size data to these for future work. We of course anticipate that appropriate use of the geographic/size data will enable stronger models, and look forward to seeing what the community can achieve using this.

---

### Decision · Program_Chairs · 2024-09-26

**Decision:**

Accept (Poster)

**Comment:**

This paper presents BIOSCAN-5M, an expanded version of its predecessor, BIOSCAN-1M. The dataset comprises over 5 million images of arthropods complemented by DNA barcode sequences, taxonomic labels, geographical information, and size information. The authors demonstrate the dataset's utility through three benchmark experiments focusing on DNA-based taxonomic classification, zero-shot image/barcode feature clustering, and multimodal retrieval across images, barcodes, and taxonomic text.

Reviewers appreciate the quality, scale, diversity, and multi-modality of the data, the evaluation of several baselines, and the clarity of the writing. Smaller concerns were raised regarding sampling bias (e.g., limited coverage in northern African and Asia) and the underutilization of certain metadata (e.g., geographical location and size). But overall I support (most) reviewers' recommendation to accept.